# AN OPERATOR NORM BASED PASSIVE FILTER PRUNING METHOD FOR EFFICIENT CNNS

## ABSTRACT

Convolutional neural networks (CNNs) have shown state-of-the-art performance in various applications. However, CNNs are resource-hungry due to their requirement of high computational complexity and memory storage. Recent efforts toward achieving computational efficiency in CNNs involve filter pruning methods that eliminate some of the filters in CNNs based on the "importance" of the filters. Existing passive filter pruning methods typically use the entry-wise norm of the filters to quantify filter importance, without considering how well the filter contributes in producing the node output. Under high pruning ratio where the large number of filters are to be pruned from the network, the entry-wise norm methods always select high entry-wise norm filters as important, and ignore the diversity learned by the other filters that may result in degradation in the performance. To address this, we present a passive filter pruning method where the filters are pruned based on their contribution in producing output by implicitly considering the operator norm of the filters. The computational cost and memory requirement is reduced significantly by eliminating filters and their corresponding feature maps from the network. Accuracy similar to the original network is recovered by fine-tuning the pruned network. The proposed pruning method gives similar or better performance than the entry-wise norm-based pruning methods at various pruning ratios. The efficacy of the proposed pruning method is evaluated on audio scene classification (e.g. TAU Urban Acoustic Scenes 2020) and image classification (MNIST handwritten digit classification, CIFAR-10).

## 1 INTRODUCTION

Convolutional neural networks (CNNs) have shown great success and exhibit state-of-the-art performance when compared to traditional hand-crafted methods in many domains (Gu et al. (2018)). Even though CNNs are highly efficient in solving non-linear complex tasks (Denton et al. (2014)), it may be challenging to deploy large-size CNNs on resource-constrained devices such as mobile phones or internet of things (IoT) devices, owing to high computational costs during inference and the memory requirement for CNNs (Simonyan & Zisserman (2015); Krizhevsky et al. (2012)). Thus, the issue of reducing the size and the computational cost of CNNs has drawn a significant amount of attention in the research community.

Recent efforts toward reducing the computational complexity of CNNs involve pruning methods where a set of parameters, such as weights or filters, are eliminated from the CNNs. These pruning methods are motivated by the existence of redundant parameters (Denil et al. (2013); Livni et al. (2014)) in CNNs that only yield extra computations without contributing much in performance (Frankle & Carbin (2019)). For example, Li et al. (2017) found that 64% of the parameters, contributing approximately 34% of the computation time, are redundant. Eliminating such redundant parameters from CNNs provides small CNNs that perform similar to the original CNNs while reducing the computations and the memory requirement compared to the original CNNs.

While eliminating weights from an unpruned CNN may result in a highly sparse network with few parameters, a pruned network obtained by eliminating individual weights is unstructured and may not be straightforward to run more efficiently. The practical acceleration in the unstructured sparse pruned networks is limited due to random connections despite high sparsity (Luo et al. (2017)). Moreover, the unstructured sparse networks can not be supported by off-the-shelf libraries and re-

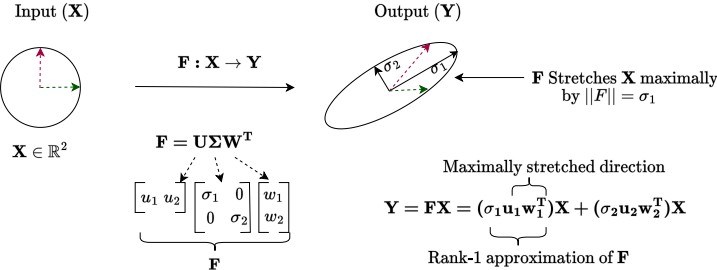

Figure 1: A geometrical view of output produced by a convolution operation, where input feature maps $\mathbf{X}$ in $\mathbb{R}^2$ are transformed to output feature maps $\mathbf{Y}$ in $\mathbb{R}^2$ using a transformation matrix $\mathbf{F}$. $\mathbf{F}$ is decomposed to a left singular matrix ($\mathbf{U}$), a right singular matrix ($\mathbf{W}$) and a diagonal matrix ($\Sigma$) using a singular value decomposition method. $\mathbf{U}$ and $\mathbf{W}$ are orthogonal matrices that cause rotation in the input, and $\sigma_1$ and $\sigma_2$ are singular values that scale the input. $\mathbf{F}$ stretches $\mathbf{X}$ maximally by $||\mathbf{F}|| = \sigma_1$ which is an operator norm of $\mathbf{F}$.

quire specialised software or hardware for speed-up (Wen et al. (2016); Han et al. (2016)). To address this unstructured pruning problem, several filter pruning methods have been proposed which eliminates whole filters, resulting in a structured pruned network (Luo et al. (2018)) that does not require additional resources for speed-up.

In these structured filter pruning methods, the "importance" of a filter, used to decide if a filter is retained or eliminated, is measured using either active or passive methods. Active filter pruning methods involve a dataset to compute the importance of filters. For example, (Luo & Wu (2017); Polyak & Wolf (2015); Hu et al. (2016); Lin et al. (2020); Yeom et al. (2021)) use feature maps which are the outputs generated from CNN filters corresponding to a set of examples, and apply metrics such as entropy, variance, average rank of feature maps and the average percentage of zeros on the feature maps to quantify the importance of the filters. Other methods including (Liu et al. (2017); Lin et al. (2019)) compute important set of filters during training process by optimizing a soft mask associated with each feature map using regularization. However, these methods are time-consuming and use extra memory resources to obtain feature maps. On the other hand, passive filter pruning methods only use the parameters of the filters without involving any dataset or optimizing process to compute the importance of the filters. Therefore, the passive filter pruning methods are less time-consuming and require less memory resources in identifying important set of filters. In particular, when there exists already a pre-trained network to be pruned, identifying pruned set of filters using optimization process (Liu et al. (2017); Lin et al. (2019)) would be heavily computationally expensive compared to that of the passive filter pruning methods.

A typical passive filter pruning method uses an entry-wise norm of the filters to measure their importance. For example, this might be a $l_1$-norm a sum of absolute values of each entry in the filter or an $l_2$-norm square root of the squared sum of each entry in the filter. Li et al. (2017) eliminates filters having smaller entry-wise $l_1$-norm or $l_2$-norm as measured from the origin, and finds that eliminating filters based on the entry-wise $l_2$-norm of the filters gives similar performance to that of the entry-wise $l_1$-norm. He et al. (2019) eliminates the filters with smaller $l_2$-norm as measured from the geometric median of all filters. Both the previous methods assume that a filter with smaller entry-wise norm is less informative or less important, without considering how significantly a filter contribute in producing output. For example, an illustration of the contribution by the filters in producing output is shown in Figure 1, where filters $\mathbf{F}$ produces an output $\mathbf{Y}$ by maximally stretching the input $\mathbf{X}$ by a largest singular value $\sigma_1$ that represents an operator norm of the $\mathbf{F}$. However, the entry-wise norm methods do not consider any input-output relationship information and rely on each entry of the filter while computing the filter importance.

To illustrate the above further, we pictorially show in Figure 2(a) that two filters $\mathbf{F}^1$ and $\mathbf{F}^3$ having same entry-wise norm contribute differently and produce different output due to different operator norm of the each filter shown in Figure 2(b). Hence such filters should be given different importance. Moreover, when a few number of filters have to be retained in the CNN to yield a very small pruned CNN, selecting filters with only high entry-wise norm may ignore the smaller norm filters that may also contribute significantly in producing output (Ye et al. (2018)). This may degrade the accuracy of

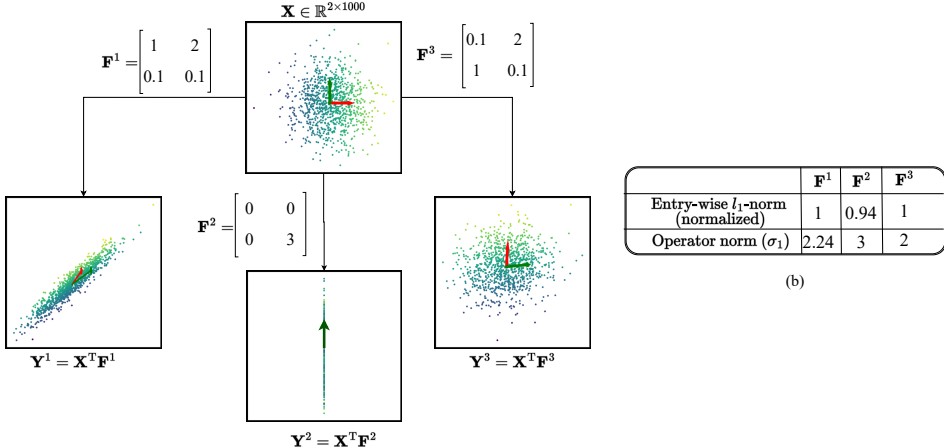

(b)

(a) Outputs produced by applying three different filters

Figure 2: (a) An illustration of output produced in the convolution layer by three CNN filters, $\mathbf{F}^1$, $\mathbf{F}^2$ and $\mathbf{F}^3$ after applying a convolution operation on $\mathbf{X} \in \mathbb{R}^{2 \times 1000}$. (b) shows the entry-wise $l_1$-norm and the operator norm of the three filters.

the pruned network significantly due to relying on the filters having high entry-wise norm, however low significance in producing output. For example as shown in Figure 2(b), $\mathbf{F}^2$ has $\sigma_1 = 3$, and it stretches $\mathbf{X}$ relatively larger than that of the $\mathbf{F}^1$ and the $\mathbf{F}^3$. However, the entry-wise norm of the $\mathbf{F}^2$ is the smallest among all three filters. Therefore $\mathbf{F}^2$ shall be ignored by the entry-wise norm methods despite a relatively high contribution than other filters.

To select the important filters based on their contribution in producing output, we propose a novel passive filter pruning framework by considering filters in a convolution layer as an operator that transform input feature maps to output feature maps. To compute the importance of filters, we use the operator norm of the filters, which represents the maximum contribution of the filters in producing output rather than relying on the entry-wise norm of the filters. Utilising all filters in a convolutional layer, we use a Rank-1 approximation of the filters to obtain the maximally stretched direction associated with the operator norm in which the input gets stretched maximally by all filters. A filter in the convolutional layer is deemed important based on how well it is aligned along the maximally stretched direction represented by all filters in that convolutional layer.

The proposed passive filter pruning method is evaluated on pre-trained CNNs such as VGGish (Hershey et al. (2017)) and DCASE 2021 Task 1A baseline (Martín-Morató et al. (2021)) designed for audio scene classification (ASC), VGG-16 network (Liu & Deng (2015))and ResNet-50 (He et al. (2016)) designed for image classification problem.

## 2 METHODOLOGY

*Notation:* Consider a CNN having $L$ convolutional layers with indexes $\in \{1, 2, \cdots, l, \cdots, L\}$. A feature map in a convolutional layer denotes an output produced by a CNN filter. In the $l^{\text{th}}$ convolution layer, let $n_l$ denote the number of input channels with each feature map of size $h_l \times w_l$. Let $\mathbf{X}$ of size $(n_{l-1} \times h_{l-1} \times w_{l-1})$ and $\mathbf{Y}$ of size $(n_l \times h_l \times w_l)$, denote the feature map matrices, produced by stacking all the respective feature maps in the $(l-1)^{\text{th}}$ and the $l^{\text{th}}$ layer respectively. A $j^{\text{th}}$ feature map in the $l^{\text{th}}$ layer is produced by a convolution operation using a $j^{\text{th}}$ filter $\mathbf{F}^{l,j}$ of size $(n_{l-1} \times k_l \times k_l)$ having $n_{l-1}$ 2D kernels each of $(k_l \times k_l)$ on $\mathbf{X}$. All filters in the $l^{\text{th}}$ convolution layer constitute a kernel tensor $\mathbf{K}_l = [\mathbf{F}^{l,1} \ \mathbf{F}^{l,2} \ \dots \ \mathbf{F}^{l,n_l}]$ of size $(n_{l-1} \times n_l \times k_l \times k_l)$. A pictorial illustration of two intermediate layers in the CNN is described in Figure 3.

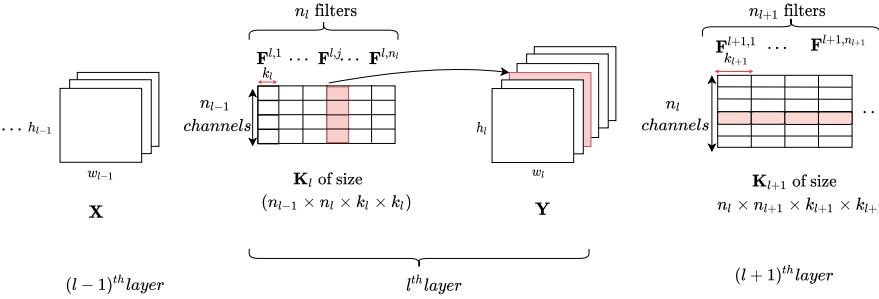

Figure 3: An illustration of the intermediate layer structure and filter pruning in CNN. In the $l^{\text{th}}$ layer, there are $n_l$ filters each having $(k_l \times k_l)$ size and $n_{l-1}$ channels. The $n_l$ filters produce $n_l$ feature maps. Pruning the $j^{\text{th}}$ filter, $\mathbf{F}^{l,j}$, in the $l^{\text{th}}$ convolution layer results in elimination of the feature map produced by the pruned filter and corresponding channel in the $(l+1)^{\text{th}}$ layer.

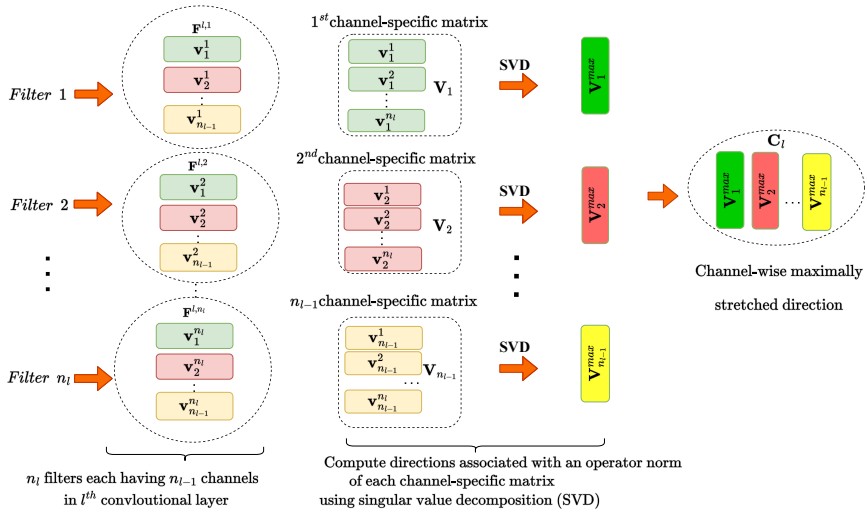

Figure 4: An illustration to obtain maximally stretched direction for each $n_{l-1}$ channels across all filters in the $l^{th}$ convolutional layer.

## 2.1 COMPUTING FILTER IMPORTANCE IN THE $l^{\text{TH}}$ CONVOLUTION LAYER

Given a kernel tensor $\mathbf{K}_l$ for the $l^{\text{th}}$ convolution layer, our aim is to compute the importance of each of the $n_l$ filters. Without loss of generality, we transform the kernel tensor $\mathbf{K}_l$ to $\mathbf{K}^{\mathbf{v}}_l$ of size $(n_{l-1} \times n_l \times k_l{}^v)$ by vectrozing the 2D kernels each of $(k_l \times k_l)$ to $k_l{}^v$.

To identify importance of each filter, we use $\mathbf{K}^{\mathbf{v}}_l$ to obtain a maximally stretched direction corresponding to the operator norm of the filters along which the filters in the given convolutional layer stretches the input maximally. Since the output produced by the $c^{\text{th}}$ channel of a filter depends only on the $c^{\text{th}}$ channel of the input in CNNs, the maximally stretched direction is obtained for each channel of the filters independently.

A channel-specific matrix $\mathbf{V}_c$ of size $(n_l \times k_l{}^v)$ is constructed by taking $c^{\text{th}}$ channel of all filters. $\mathbf{V}_c$ denotes the learned weights of the $c^{\text{th}}$ channel across the $n_l$ filters. Next, singular value decomposition (SVD) is performed on $\mathbf{V}_c$ to compute a rank-1 approximation of the $c^{\text{th}}$ channel as $\mathbf{V}_c \approx \sigma_1 \mathbf{u}_1 \mathbf{w}_1^{\text{T}}$. Here, $\sigma_1$ denotes the maximum singular value that affects the corresponding channel of the input maximally and is equivalent to the operator norm of the $\mathbf{V}_c$, $\mathbf{u}_1$ is the first left singular vector, and $\mathbf{w}_1$ is the first right singular vector. A row of $\mathbf{V}_c$ normalized to unit norm is denoted as $\mathbf{V}_c^{\text{max}}$, and is considered as a maximally stretched direction for the $c^{\text{th}}$ channel. $\mathbf{V}_c^{\text{max}}$ provides a reference for measuring the significance the $c^{\text{th}}$ channel of the filter.

---

**Algorithm 1** Filter Importance Calculation

---

**input** : $\mathbf{K^v}_l$ of size $(n_{l-1} \times n_l \times k_l{}^v)$, kernel matrix in $l^{\text{th}}$ layer .
**output:** Score_norm, # normalized importance scores of filters, in $l^{\text{th}}$ layer.
Initialization: $\mathbf{C}_l$ = [ ], Score = [ ], # importance score of filters.
  ... # Obtaining channel-wise maximally stretched direction associated with operator norm...
  **for** $c \leftarrow 1$ **to** $n_{l-1}$ **do**
    $\mathbf{V}_c$ =$\mathbf{K^v}_l$[c , : , :] # Take $c^{\text{th}}$ channel of size $(n_l \times k_l{}^v)$ from all the filters,
    $\mathbf{u}, \Sigma, \mathbf{w}$ = SVD( $\mathbf{V}_c$ ) # Perform SVD on $\mathbf{V}_c$,
    $\mathbf{C}_l$.append(($\mathbf{u}_1\mathbf{w}_1^{\text{T}}$)[1,:]) #Take first left ($\mathbf{u}_1$) and first right ($\mathbf{w}_1$) singular vector associated
    with $\sigma_1$, take any row of $\mathbf{u}_1\mathbf{w}_1^T$ and normalized to 1 and append the normalized row to $\mathbf{C}_l$.
**end**
..................# Importance-score calculation...................
  **for** $j \leftarrow 1$ **to** $n_l$ **do**
    $\mathbf{F}^{l,j}$ = $\mathbf{K^v}_l$[ : , j , :]   # Take $j^{\text{th}}$ filter.
    $\bar{\mathbf{F}}^{l,j}$= Score.append([trace(($\mathbf{F}^{l,j})\mathbf{C}_l$)]) #Compute importance.
**end**
$\alpha$ =[Score]
  Score_norm = $\frac{\alpha^2}{\max(\alpha^2)}$ # Normalized importance.
return Score_norm

---

After obtaining $\mathbf{V}_c^{\text{max}}$ corresponding to the $c^{\text{th}}$ channel, we obtain maximally stretched directions for other channels. Finally, the $\bar{\mathbf{V}}_c^{\text{max}}$ of all channels are stacked together to yield $\mathbf{C}_l$ of size $(k_l{}^v \times n_{l-1})$ for a given convolutional layer. A pictorial representation to obtain $\mathbf{C}_l$ is shown in Figure 4.

Given $\mathbf{C}_l$, the importance for the $j^{\text{th}}$ filter is computed as, $\text{Trace}[(\mathbf{F}^{l,j})\mathbf{C}_l)]$. The filters are ranked as per their importance with a relatively high importance score indicates a high contribution of the filter in producing output. Algorithm 1 summarizes the process to compute the importance of various filters in a given convolutional layer.

After ranking the filters based on their importance for various convolutional layers, few unimportant filters are eliminated based on a user-defined pruning ratio for various convolutional layers of CNN, and a pruned network is obtained. Pruning a filter in the $l^{\text{th}}$ convolution layer also eliminates the feature map produced by the pruned filter and the related kernel or channel in the next layer as shown in Figure 3, hence the computations is reduced in both the $l^{\text{th}}$ layer and the $(l+1)^{\text{th}}$ layer. In the end, a fine-tuning of the pruned network is performed to regain the performance loss due to elimination of filters from the original network.

## 3 EXPERIMENTS

We evaluate the proposed pruning method on CNNs designed for audio scene classification (ASC) and image classification. A brief summary of the unpruned CNNs used for experimentation is shown in Figure 5 and is described below,

**Unpruned CNNs:** We use two different unpruned networks for ASC, (a) VGGish_Net and (b) DCASE21_Net. We also use (c) VGG-16 network for image classification.

(a) **VGGish_Net:** The VGGish_Net is built using a publicly available pre-trained VGGish network (Hershey et al. (2017)) followed by a dense and a classification layer. We train the VGGish_Net on TUT Urban Acoustic Scenes 2018 development (we denote it as DCASE-18) training dataset (Mesaros et al. (2018)) to classify 10 different audio scenes using Adam optimizer with cross-entropy loss function for 200 epochs. The input to the VGGish_Net is a log-melspectrogram of size $(96 \times 64)$ computed corresponding to 960ms audio segment for a whole 10s audio scene. The VGGish_Net has 55,361,162 (approximately 55M) parameters, requiring 903M multiply-accumulate operations (MACs)[1] during inference corresponding to an audio clip of 960ms and gives 64.69% accuracy on 10s audio scene for

---

[1]To compute MACs, we use publicly available "nessi.py" script available at https://github.com/AlbertoAncilotto/NeSsi

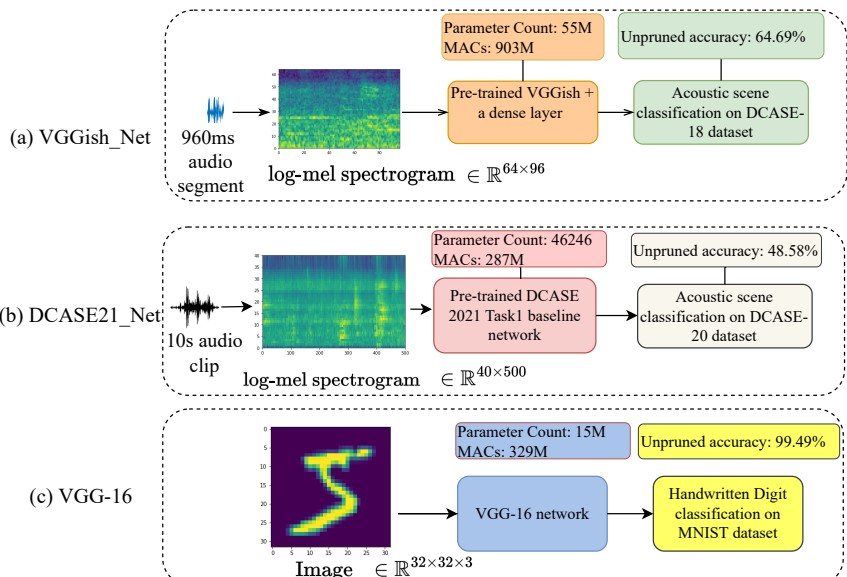

Figure 5: Unpruned CNNs used for experimentation; (a) VGGish_Net (b) DCASE21_Net and (c) VGG_Net.

DCASE-18 development validation dataset. The VGGish_Net has six convolution layers (termed as C1 to C6).

(b) **DCASE21_Net:** DCASE21_Net is a publicly available pre-trained network designed for DCASE 2021 Task 1A that is trained using TAU Urban Acoustic Scenes 2020 Mobile development dataset (we denote "DCASE-20") to classify 10 different audio scenes (Martín-Morató et al. (2021)). The input to the network is a log-melspectrogram of size ($40 \times 500$) corresponding to a 10s audio clip. The trained network has 46246 parameters, requiring approximately 287M MACs during inference corresponding to 10-second-length audio clip and gives 48.58% accuracy on the DCASE-20 development validation dataset. DCASE21_Net consists of three convolutional layers (termed as C1 to C3) and one fully connected layer.

(c) **VGG-16:** We use a VGG-16 network ( (Liu & Deng (2015))) for handwritten digit classification using MNIST dataset[2](LeCun et al. (2010)). We train the VGG-16 from scratch for 200 epochs on the MNIST training dataset. The VGG-16 consists of 13 convolutional layer (termed as C1 to C13) and 2 dense layers. The VGG-16 has 15M parameters and requires 329M MACs per inference and the trained VGG-16 gives 99.49% accuracy for the MNIST testing dataset.

Apart from unpruned networks (a), (b) and (c), we evaluate the prposed method on VGG-16 and ResNet-50 (He et al. (2016)) networks using CIFAR-10 dataset (Krizhevsky & Hinton (2009)) and an initial analysis on ResNet-50 using CIFAR-10 dataset. Please refer Appendix for detailed analysis.

**Fine-tuning of the pruned network:** After computing importance of the filters using Algorithm 1 across various convolutional layers, we eliminate $p$ percentage of unimportant filters from various convolutional layer, where $p \in \{25, 50, 75, 90\}$ denotes a pruning ratio. Once the pruned network is obtained, we re-train the network with similar conditions such as same optimizer as used in the training of the unpruned networks (a), (b) and (c) for 100 epochs. The codes for the proposed pruning framework, various pruned and unpruned networks can be found at the link[3].

---

[2]The MNIST dataset is downloaded using Keras API which has a pre-defined training and testing set of grayscale images. Each grayscale image of size ($28 \times 28$) is converted into three channels by depthwise stacking the grayscale image and then reshaping to ($32 \times 32 \times 3$) that is used as an input to VGG-16.

[3]https://anonymous.4open.science/r/Operator_norm_passive_Filter_Pruning-125D

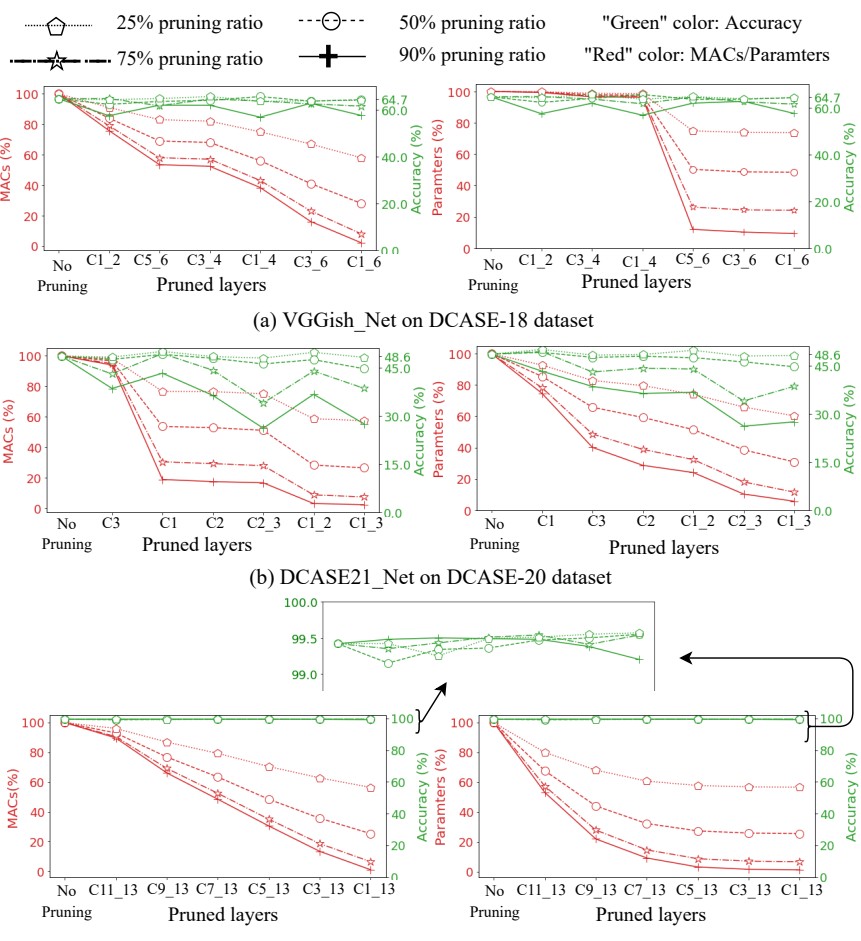

(a) VGGish_Net on DCASE-18 dataset

(b) DCASE21_Net on DCASE-20 dataset

(c) VGG-16 on MNIST dataset

Figure 6: Accuracy, MACs and parameters across different pruned networks for (a) VGGish_Net, (b) DCASE21_Net and (c) VGG-16 network, when different subsets of convolutional layers are pruned at different pruning ratios. Here, "CA_B" as "pruned layers" means that convolution layers from A to B are pruned.

**Other methods for comparison:** We compare the proposed operator norm based pruning method with that of the entry-wise norm based methods, (a) $l_1$-norm method that eliminates filters with smaller entry-wise $l_1$-norm (Li et al. (2017)) and (b) geometric median (GM) method that eliminates filters with smaller $l_2$-norm as measured from the geometric median of all filters (He et al. (2019)). We also compare the proposed pruning method with the existing active filter pruning methods including HRank (Lin et al. (2020)), Energy-aware pruning (Yeom et al. (2021)), L1-slimming (Liu et al. (2017)), and GAL-0.1 (Lin et al. (2019)) in Appendix: Table 3.

## 4 RESULTS

We analyse accuracy, the number of MACs and the number of parameters in the pruned networks obtained after pruning various subsets of convolution layers at different pruning ratios from the unpruned network.

**VGGish_Net and DCASE21_Net on ASC:** As shown in Figure 6 (a,b), the accuracy obtained using the various pruned networks is similar to that of the unpruned networks at $p = 25\%$. For VG-Gish_Net, the number of MACs are reduced by 40 percentage points and the parameters are reduced by 25 percentage points, when all convolutional layers (C1_6: C1 to C6) are pruned at $p = 25\%$ shown in Figure 6(a). For DCASE21_Net, both the MACs and the parameters are reduced by 40

Table 1: Comparison of accuracy, the MACs and the parameters among the (a) unpruned, (b) the pruned network obtained using the proposed pruning method and the same pruned network obtained in (b), however, trained from scratch.

| Network | Accuracy (%) | MACs | Reduced MACs | Paramters | Reduced paramters |
|---|---|---|---|---|---|
| VGGish_Net | 64.69 | 903M | - | 55M | - |
| (a) Pruned VGGish_Net | 64.30 | 253M | 72% | 26.82M | 52% |
| (b) Pruned VGGish_Net-scratch | 61.40 | | | | |
| DCASE21_Net | 48.58 | 286M | - | 46246 | - |
| (a) Pruned DCASE21_Net | 48.18 | 164M | 43% | 27906 | 40% |
| (b) Pruned DCASE21_Net-scratch | 46.80 | | | | |
| VGGNet-MNIST | 99.49 | 329M | - | 15M | - |
| (a) Pruned VGGNet-MNIST | 99.10 | 3.33M | 99% | 0.18M | 99% |
| (b) Pruned VGGNet-MNIST-scratch | 98.97 | | | | |

percentage points when all convolutional layers (C1_3: C1 to C3) are pruned at $p = 25\%$ shown in Figure 6(b).

At $p = 50\%$, the accuracy drop across various pruned networks is less than 4 percentage points compared to that of the unpruned network for both VGGish_Net and DCASE21_Net as shown in Figure 6(a, b). For VGGish_Net, the number of MACs are reduced by 75 percentage points and the parameters are reduced by 55 percentage points, when all convolutional layers (C1_6: C1 to C6) are pruned at $p = 50\%$ shown in Figure 6(a). For DCASE21_Net, both the MACs and the parameters are reduced by 75 percentage points when all convolutional layers (C1_3: C1 to C3) are pruned at $p = 50\%$ shown in Figure 6(b).

At $p = 75\%$, the accuracy drop across various pruned networks is less than 5 percentage points and 10 percentage points compared to that of the unpruned network for both VGGish_Net and DCASE21_Net respectively as shown in Figure 6(a, b). The accuracy of the pruned networks degrades further at $p = 90\%$ and the drop in accuracy for VGGish_Net and DCASE21_Net is 10 percentage points and 20 percentage points respectively, when all layer are pruned at $p = 90\%$. On the other hand, both the MACs and the parameters are reduced significantly by more than 75 percentage points when all convolutional layers of VGGish_Net and DCASE21_Net are pruned at $p = \{75\%, 90\%\}$ as shown in Figure 6(a, b).

In general, the MACs, the parameters and the accuracy decrease when various convolutional layers are pruned from 25% to 90% pruning ratio. The accuracy of the pruned DCASE21_Net reduces significantly from 0.2 percentage points to 20 percentage points compared to that of the unpruned network, when all layers are pruned with 25% to 90% pruning ratio. This might be due to the smaller network size of the DCASE21_Net, where eliminating large number of parameters at high pruning ratio results in under-fitting problem due to insufficient parameters.

**VGG-16 for image classification :** For VGG-16, the accuracy of the various pruned networks is reduced by less than 0.5% percentage points compared to that of the unpruned VGG-16 as shown in Figure 6(c). The number of MACs and the parameters are reduced from approximately 40 percentage points to 99 percentage points when pruning ratio across all convolutional layers (C1_13: C1 to C13) varies from 25% to 90% respectively.

Next, Table 1 compares the accuracy, MACs and parameters in the unpruned network with that of (a) the pruned network having important filters obtained using the proposed pruning method as an initialisation before fine-tuning process, (b) the same pruned network as obtained in (a) and trained from scratch with random initialization. For comparison, we choose the pruned network in (a) which gives a maximum drop of 0.5 percentage point in accuracy compared to that of the unpruned network. For various networks, the parameters and the MACs are reduced significantly with a marginal drop in accuracy compared to that of the unpruned network. The accuracy of (a) the pruned network obtained from the proposed pruning method is better than (b) the same pruned network trained from scratch. This shows that it is beneficial to train a large network, and then perform pruning to reduce the parameters rather than training the same size pruned network from scratch. For experimental analysis of VGG-16 network on CIFAR-10 and ResNet-50 network on CIFAR-10, please refer to Appendix A.1.2 and Appendix A.2.2 respectively.

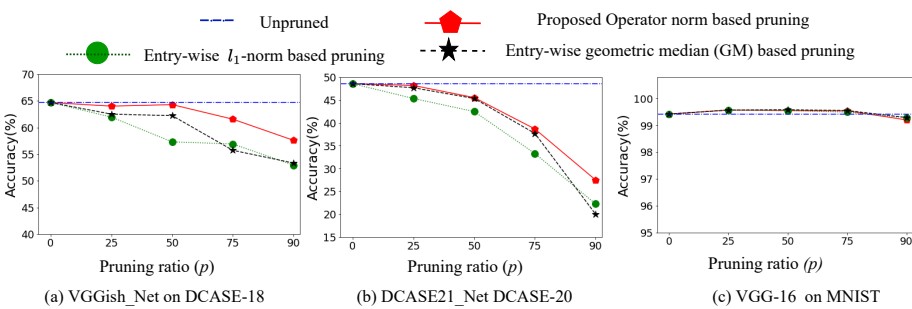

Figure 7: Accuracy obtained using the pruned network when all layers are pruned at different pruning ratio using entry-wise $l_1$-norm method, geometrical-median (GM) method and the proposed operator norm pruning method across various networks.

**Comparison with other methods:** Figure 7 compares the accuracy of the proposed operator norm pruning method with that of the entry-wise $l_1$-norm and the geometrical median (GM) based methods, when all layers in the unpruned network are pruned at different pruning ratios. For VGGish_Net and DCASE21_Net, the accuracy obtained using the proposed operator norm based pruning method is better than that of the entry-wise norm methods particularly when a large number of filters ($p = 90\%$) are pruned from the network as shown in Figure 7(a, b). Moreover, the pruned network obtained using the proposed operator norm based pruning method recovers faster to regain the accuracy during the fine-tuning process compared to that of the other pruning methods as shown in Appendix: Figure 8(a, b) at different pruning ratios. For VGG-16, the accuracy obtained using the proposed operator norm pruning method is similar to the other methods as shown in Figure 7(c) and recovers faster compared to that of the other pruning methods as shown in Appendix: Figure 8(c) at different pruning ratios.

For VGG-16 on CIFAR-10, we find that the accuracy obtained using the proposed pruning method is slightly better than that of the entry-wise norm methods at various pruning ratio (Appendix: Figure 10), and the convergence speed is also more or less similar to the entry-wise norm methods (Appendix: Figure 11). Similar analysis is also observed for ResNet-50 network on CIFAR-10 dataset (Appendix: Figure 15). This suggests that the proposed pruning method performs similar to or better than the existing entry-wise norm based pruning methods across different CNNs designed for audio and image classification , hence showing better generalization. This shows that selecting important filters based on the operator norm of the filters in producing output is advantageous compared to considering entry-wise norm of the filters due to similar or better performance across various CNNs.

In comparison to the active filter pruning methods including HRank (Lin et al. (2020)), Energy-aware pruning (Yeom et al. (2021)), L1-slimming (Liu et al. (2017)), and GAL-0.1 (Lin et al. (2019)) as given in Appendix: Table 3, the proposed pruning method reduces similar number of parameters and can achieve accuracy within $\approx 1$ percentage point as given by the active filter pruning methods, with an advantage of no requirement of dataset, feature maps, regularization techniques or knowledge distillation techniques while identifying pruned set of filters.

## 5 CONCLUSION

This paper presents a passive filter pruning method which reduces the computational complexity and the memory of the unpruned CNN significantly at marginal drop in accuracy. The proposed pruning method reduces similar number of parameters and MACs at a maximum drop in accuracy of less than 1 percentage points compared to the existing active filter pruning methods without involving any dataset or regularization methods while eliminating filters from the unpruned network. We find that the proposed pruning method gives similar or better accuracy at various pruning ratio compared to that of the existing entry-wise norm methods across different CNNs designed in audio and image domains, hence showing the generalization ability across different domains as well. In future, we would like to improve the performance of the pruned network by incorporating data augmentation techniques to improve the fine-tuning process.

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

# A APPENDIX

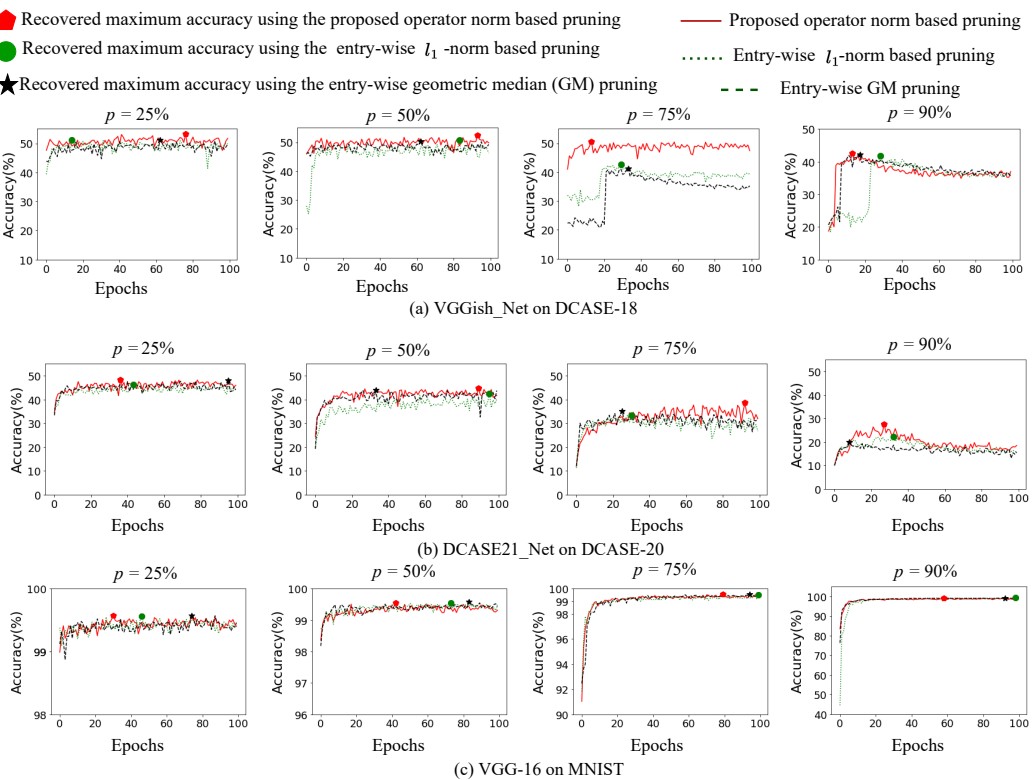

Figure 8: Convergence plots showing regain in the accuracy during fine-tuning of the pruned network when the pruned network is obtained by pruning all convolutional layers at different pruning ratios.

## A.1 EXPERIMENTAL SETUP AND ANALYSIS: VGG-16 ON CIFAR-10

### A.1.1 EXPERIMENTAL SETUP

**Unpruned network:** We use a publicly available pre-trained VGG-16 network trained on CIFAR-10 dataset (Geifman) as an unpruned network. The unpruned architecture is based on the VGG-16 (Simonyan & Zisserman (2015)) with adaptation to CIFAR-10 dataset based on (Liu & Deng (2015)). The VGG-16 has 15M parameters and requires 329M MACs per inference and the pre-trained VGG-16 gives 93.58% accuracy for the CIFAR-10 testing dataset. The unpruned network has 13 convolutional layer, denoted as C1 to C13.

To perform fine-tuning of the pruned network, we re-train the pruned network for 100 epochs with similar conditions such as same optimizer as used in training the unpruned network (Geifman).

**Other methods used for comparison:** For comparison, we use passive filter pruning methods such as entry-wise $l_1$-norm (Li et al. (2017)) and entry-wise geometric median (GM) (He et al. (2019)). We also compare the proposed pruning method with the existing active filter pruning methods including HRank (Lin et al. (2020)), Energy-aware pruning (Yeom et al. (2021)), L1-slimming (Liu et al. (2017)), and GAL-0.1 (Lin et al. (2019)). A brief overview of the active filter pruning methods used for comparison is given below,

**HRank:** This method opts three steps to obtain a pruned network. 1) A set of feature maps are generated for a given filter using a set of examples. 2) Rank of the feature maps is computed and an average rank computed across feature maps of various examples is used as a criterion to quantify

filter importance. 3) The filters with low average rank are eliminated and a fine-tuning procedure is opted to compensate the performance loss.

**Energy-aware:** This method uses a set of input data to generate feature maps for a given convolutional layer and then compute energy of each feature map by computing nuclear norm (sum of all singular values) of each feature map. A feature map with low energy is pruned and then fine-tuning of the pruned network is performed.

**L1-slimming**: This method uses three steps to obtain a pruned network: (1) Trains the underlying unpruned network with channel-level sparsity regularisation, (2) Eliminate the channels (feature maps and corresponding filters) having small value of a scaling coefficient ($\gamma$) in a batch normalisation layer, and (3) Fine-tune the pruned network.

**GAL-0.1:** This method applies knowledge distillation to train the pruned network with $l_1$-regularisation on the soft mask associated with channels (feature maps) to mimic the unpruned network by aligning their output using generative adversarial learning (GAL) from the unpruned network to the pruned network. Subsequently, the pruned network is fine-tuned after eliminating the unimportant feature maps and corresponding filters.

### A.1.2 PERFORMANCE ANALYSIS

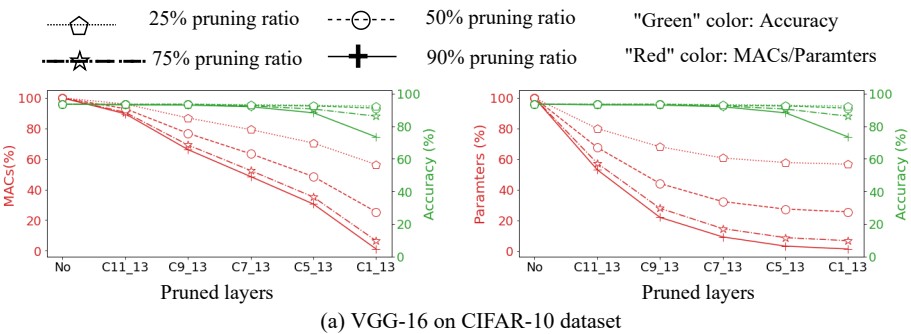

(a) VGG-16 on CIFAR-10 dataset

Figure 9: Accuracy, MACs and parameters across different pruned networks for VGG-16 network trained on CIFAR-10, when different subsets of convolutional layers are pruned at different pruning ratios. Here, "CA_B" as "pruned layers" means that convolution layers from A to B are pruned.

Figure 9 shows the number of MACs and the number of parameters in the pruned network obtained after pruning various subsets of convolution layers at different pruning ratios from the unpruned network. By varying $p$ from 25% to 70% across convolution layers from C5 to C13, the number of parameters are reduced from 40% to 90%, and the MACs are reduced from 25% to 70% with an accuracy drop from 0.25 to 3 percentage points respectively. Pruning all convolutional layers at $p = 25\%$ results in approximately 45% reduction in noth parameters and MACs at marginal drop in accuracy compared to that of the unpruned network. On the other hand, pruning all convolutional layers at $p = 90\%$ reduces parameters and MACs significantly, however accuracy drop is approximately 20 percentage points compared to the unpruned network. Therefore, the pruning ratio can be chosen according to the requirement whether the underlying resources (computation and parameters) are primarily important or the accuracy.

Next, Table 2 compares the accuracy, MACs and parameters in the unpruned network with that of (a) the pruned network having important filters obtained using the proposed pruning method as an initialisation before fine-tuning process, (b) the same pruned network as obtained in (a) and trained from scratch with random initialization. For comparison, we choose the pruned network in (a) which gives a maximum drop of approximately 0.5 percentage point in accuracy compared to that of the unpruned network. The parameters and the MACs are reduced by 34% and 78% respectively with a marginal drop in accuracy compared to that of the unpruned network. The accuracy of (a) the pruned network obtained from the proposed pruning method is better than (b) the same pruned network trained from scratch.

Table 2: Comparison of accuracy, the MACs and the parameters among the (a) unpruned, (b) the pruned network obtained using the proposed pruning method and the same pruned network obtained in (b), however, trained from scratch.

| Network | Accuracy (%) | MACs | Reduced MACs | Parameters | Reduced parameters |
|---|---|---|---|---|---|
| VGG-16 on CIFAR10 | 93.56 | 329M | - | 15M | - |
| (a) Pruned VGG-16 on CIFAR10 | 93.00 | 217M | 34% | 3.29M | 78% |
| (b) Pruned VGG-16 on CIFAR10-scratch | 86.02 | | | | |

**Comparison with entry-wise norm methods:** Figure 10 and Figure 11 compares accuracy and convergence during fine-tuning of the pruned network among the proposed pruning method, entry-wise $l_1$-norm and entry-wise GM pruning methods at different pruning ratio. In general, we find that the accuracy obtained using the proposed pruning method is similar or marginally better (0.5 to 1 percentage points) than that of the entry-wise norm methods particularly at high pruning ratio as shown in Figure 10 with similar convergence speed ($\pm$ 0 to 20 epochs) in recovering the accuracy during fine-tuning as shown in Figure 11.

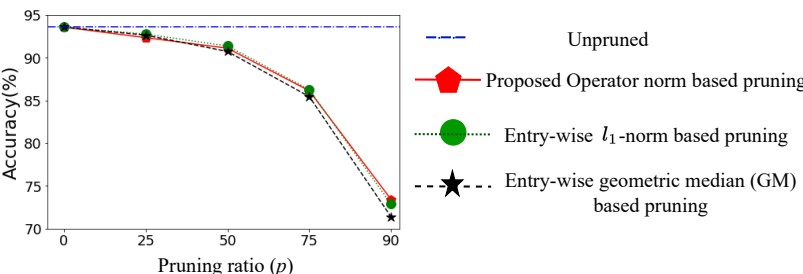

Figure 10: Accuracy obtained using the pruned network when each layer of VGG-16 network on CIFAR-10 dataset is pruned at $p$ pruning ratio using entry-wise $l_1$-norm method, geometrical-median (GM) method and the proposed operator norm pruning method.

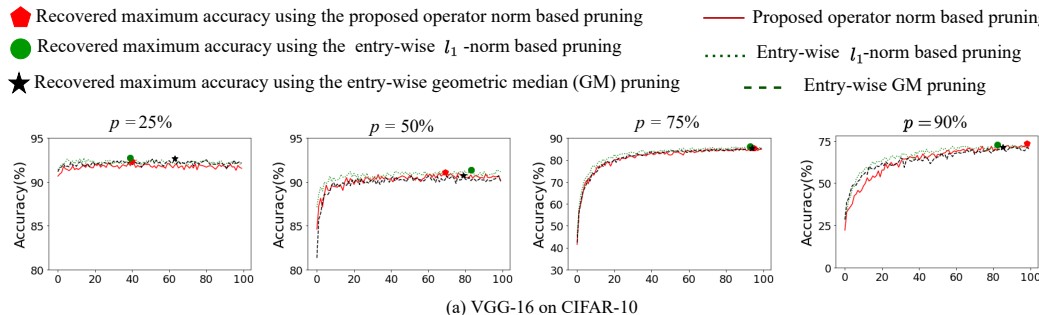

Figure 11: Convergence plots showing regain in the accuracy during fine-tuning of the pruned VGG-16 network on CIFAR-10 when the pruned network is obtained by pruning each convolutional layers at $p$ pruning ratios.

**Comparison with existing active filter pruning methods:** Table 3 compares the proposed pruning method with existing active filter pruning methods. In general, pruning filters from the unpruned network using only filters result in drop in accuracy approximately 1 percentage points compared to pruning filters by involving a dataset, using feature maps, regularization or knowledge distillation (KD) at similar reduction in parameters.

In comparison to feature map based pruning methods such as HRank and Energy-aware as given in Table 3, we find that the proposed filter based pruning method can achieve similar performance compared to that of feature map based pruning method, however the proposed method requires more number of parameters and MACs.

Table 3: Comparison of accuracy, the MACs and the parameters among different existing methods for VGG-16 on CIFAR-10 dataset. Please note that the unpruned network used for each comparison methods has different training procedure and unpruned accuracy. Therefore, we compare the parameters, MACs and accuracy obtained from the pruned network obtained from each method. Here "FT" is fine-tuning, "KD" is knowledge distillation.

| Method | Data used in Pruning | KD | FT | Accuracy (%) | Parameters | MACs |
|---|---|---|---|---|---|---|
| HRank (Lin et al. (2020)) | ✓ | ✗ | ✓ | 93.43 | 2.50M | 146M |
| Ours ($p = 90\%$, C7_13) | ✗ | ✗ | ✓ | 92.00 | 1.37M | 159M |
| Ours ($p = 75\%$, C7_13) | ✗ | ✗ | ✓ | 92.46 | 2.18M | 172M |
| Ours ($p = 75\%$, C9_13) | ✗ | ✗ | ✓ | 93.25 | 4.18M | 227M |
| Ours ($p = 50\%$, C9_13) | ✗ | ✗ | ✓ | 93.50 | 6.60M | 251M |
| Energy-aware pruning (Yeom et al. (2021)) | ✓ | ✗ | ✓ | 93.48 | 2.86M | 104.67M |
| Ours ($p = 75\%$, C7_13) | ✗ | ✗ | ✓ | 92.46 | 2.18M | 172M |
| Ours ($p = 50\%$, C9_13) | ✗ | ✗ | ✓ | 93.50 | 6.60M | 251M |
| L1-slimming (Liu et al. (2017)) | ✓ | ✗ | ✓ | 93.80 | 2.30M | 391M |
| Ours ($p = 50\%$, C9_13) | ✗ | ✗ | ✓ | 93.50 | 6.60M | 251M |
| GAL-0.1 (Lin et al. (2019)) | ✓ | ✓ | ✗ | 90.78 | 2.67M | 171.89M |
| GAL-0.1 | ✓ | ✓ | ✓ | 93.42 | 2.67M | 171.89M |
| ours ($p = 75\%$, C7_13) | ✗ | ✗ | ✓ | 92.46 | 2.18M | 172.29M |

In comparison to L1-slimming method as given in Table 3, the proposed pruning method results in similar accuracy and requires 1.6 times less MACs, however with 3 times more requirement of parameters. The L1-slimming method prunes filter based on $\gamma$ parameters in the batch normalisation layer and hence, solely dependent on batch normalisation layer. Moreover, the L1-Slimming method requires training of the unpruned network to identify which feature maps need to be pruned. Assuming that there exists already a pre-trained network which is to be pruned, the L1-slimming method would require extra computations to obtain the pruned network. On the other hand, the proposed pruning method can directly use pre-trained filters to obtain the pruned network. Although, the proposed pruning method result in 3 times more parameters that require more memory size, yet the quantization (float32 to INT8) of the pruned network can be performed to reduce the pruned network size by 4 times at marginal ($< 0.15$ percentage point) drop in accuracy. Therefore, the proposed pruning method is advantageous over the L1-slimming pruning method providing similar accuracy at reduced MACs.

The GAL-0.1 (without fine-tuning (FT)) method is closely related to the proposed pruning method, where knowledge distillation step to train the pruned network can be considered as equivalent to the fine-tuning of the pruned network in the proposed pruning method. In contrast to GAL-0.1 without FT, the proposed pruning method gives better performance as given in Table 3. However, fine-tuning the pruned network obtained using (GAL-0.1 without FT) results in approximately 1 percentage points better performance compared to that of the proposed pruning method.

The proposed pruning method is simple and advantageous in terms of obtaining pruned network using filters only compared to involving dataset based pruning methods which results in 1 percentage points better accuracy, however with relatively more efforts, resources (memory, extra parameters such as learning soft mask) compared to the proposed pruning method.

## A.2 EXPERIMENTAL SETUP AND ANALYSIS: RESNET-50 ON CIFAR-10

### A.2.1 EXPERIMENTAL SETUP

**Unpruned network:** We perform a preliminary analysis on ResNet-50 using CIFAR-10 dataset. We use a pre-trained ResNet-50 with ImageNet (He et al. (2016)) weights followed by a global average pooling, a fully connected layer and a classification layer to train an unpruned network for CIFAR-10. The architecture used for experiments is shown in Figure 12 consisting of various stages and different blocks. The unpruned network is trained for 300 epochs with stochastic gradient descent (SGD) optimizer and cross entropy loss function. The unpruned network has 24M parameters, 99M MACs andd gives 83.37% accuracy on CIFAR-10 test dataset with (32 x 32 x 3) shaped input.

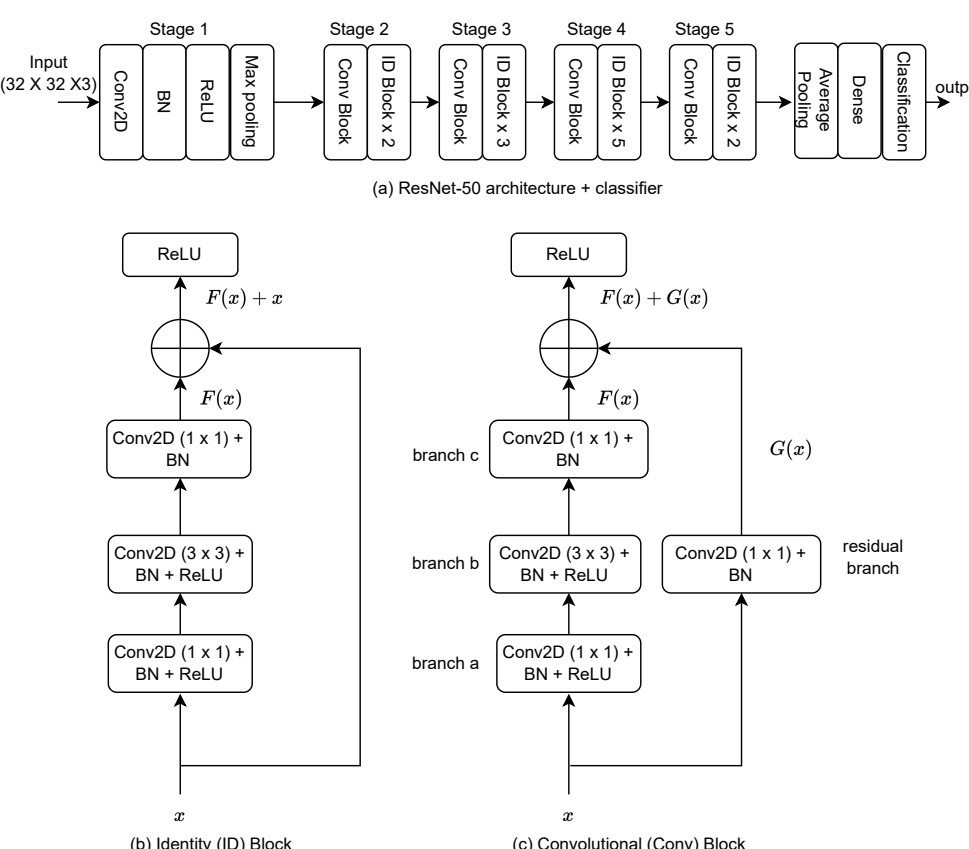

Figure 12: (a) Architecture to classify CIFAR-10 dataset using ResNet-50 (stage 1 to stage 5) which comprises of (b) identity blocks and (c) convolutional blocks, followed by global average pooling layer, a dense and a classification layer. "BN" denotes batch normalization. "ReLU" denotes recitified linear unit activation function and Conv2D is a convolutional layer with 2D filters.

**Pruning layers in ResNet-50:** In the current preliminary analysis for ResNet-50, we consider convolutional layers having (3 x 3) filters for pruning since they contain more parameters and computations compared to that of the convolutional layers with (1 x 1) filters. The proposed pruning method can be applied directly to the convolutional layers in the residual branch as well. For this, the same number of filters from the convolutional layer in the residual branch should be pruned as that of the convolutional layer from the main "branch c" as shown in Figure 12(c). This is to ensure the same dimensionality while performing addition operation between the main branch output $F(x)$ and the residual branch output $G(x)$.

We analyse performance of the pruned network when various convolutional layers with (3 x 3) filters are pruned from (a) stage 1 to stage 5, (b) stage 2 to stage 5 and (c) stage 3 to stage 5 at different pruning ratios.

### A.2.2 PERFORMANCE ANALYSIS

Figure 13 shows accuracy obtained at different pruning ratio after pruning convolutional layers from different stages in ResNet-50. Pruning convolutional layers from stage 1 to stage 5 result in more reduction in accuracy compared to that of stage 3 to stage 5 at various pruning ratios. Pruning convolutional layers from stage 3 to stage 5 result in accuracy drop from -0.15 to 2.5 percentage points, reduces parameters from 17% to 60% and reduces MACs from 14% to 54%, when $p$ is varied from 25% to 90% as showing in Figure 14.

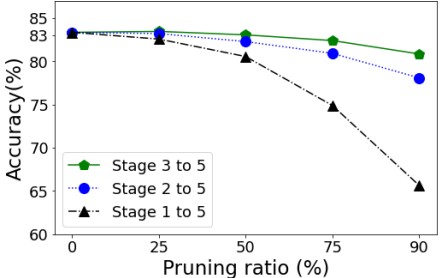

Figure 13: Accuracy when each convolutional layer having (3 x 3) filters from various stages in ResNet-50 are pruned at different pruning ratio.

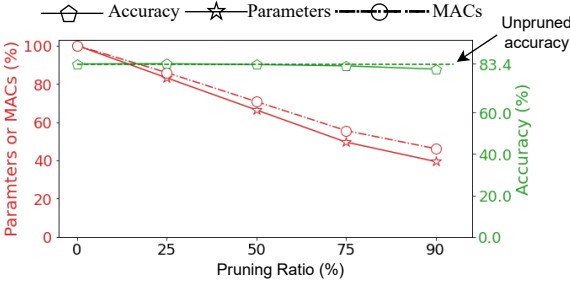

Figure 14: Accuracy, MACs and parameters obtained after pruning each convolutional layers having (3 x 3) filters from stage 3 to stage 5 at different pruning ratio.

Next, Table 4 compares the accuracy, MACs and parameters in the unpruned network with that of (a) the pruned network having important filters obtained using the proposed pruning method as an initialisation before fine-tuning process, (b) the same pruned network as obtained in (a) and trained from scratch with random initialization. For comparison, we choose the pruned network in (a) which gives a maximum drop of approximately 0.5 percentage point in accuracy compared to that of the unpruned network. The parameters and the MACs are reduced by 33% and 30% respectively with a marginal drop in accuracy compared to that of the unpruned network. The accuracy of (a) the pruned network obtained from the proposed pruning method is significantly better than (b) the same pruned network trained from scratch. This suggests that it is useful to obtain a smaller size network from a relatively large-size network compared to obtaining similar small architecture trained from scratch.

Table 4: Comparison of accuracy, the MACs and the parameters among the (a) unpruned, (b) the pruned network obtained using the proposed pruning method and the same pruned network obtained in (b), however, trained from scratch.

| Network | Accuracy (%) | MACs | Reduced MACs | Parameters | Reduced parameters |
|---|---|---|---|---|---|
| ResNet-50 on CIFAR10 | 83.37 | 99M | - | 24M | - |
| (a) Pruned ResNet-50 on CIFAR10 | 83.08 | 70M | 30% | 16M | 33% |
| (b) Pruned ResNet-50 on CIFAR10-scratch | 66.45 | | | | |

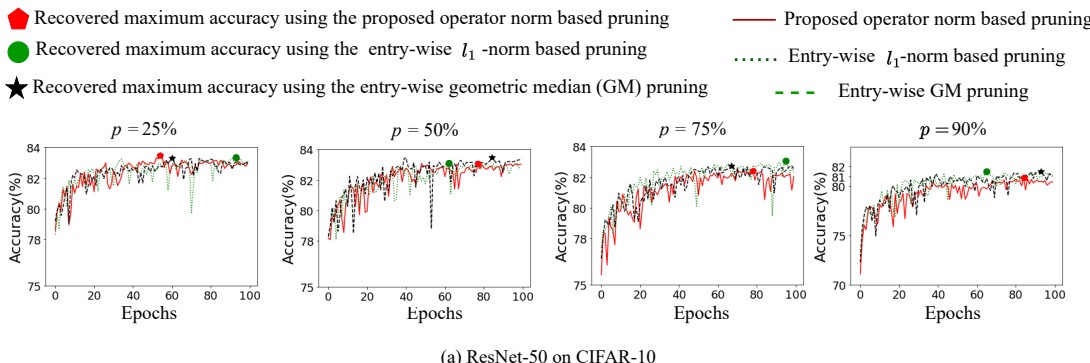

(a) ResNet-50 on CIFAR-10

Figure 15: Convergence plot during fine-tuning process of pruned ResNet-50 (stage3 to stage 5) on CIFAR-10 at different pruning ratio.

In comparison to existing passive filter pruning methods such as $l_1$-norm and geometric median (GM), the proposed pruning method gives similar performance ($\pm$ 0.5 percentage points) and recovers accuracy at similar epochs ($\pm$ 20) at different pruning ratio as shown in Figure 15.

