# OpenReview forum: "AN OPERATOR NORM BASED PASSIVE FILTER PRUNING METHOD FOR EFFICIENT CNNS"
_ICLR.cc/2023/Conference — Submitted to ICLR 2023_

### Official Review · Reviewer_mkSs · 2022-10-20

**Confidence:** 3
**Correctness:** 3
**Technical Novelty And Significance:** 2
**Empirical Novelty And Significance:** 2
**Recommendation:** 5

**Clarity, Quality, Novelty And Reproducibility:**

- This paper is clear and easy to understand because figure(s) summarizes the contents of the paper well.
- This paper used SVD for passive filter pruning without dataset intervention. However, since SVD is already widely used in the pruning field, it seems insufficient in terms of novelty.

**Strength And Weaknesses:**

Strength
- The paper is written clearly and easy to understand.

Weaknesses
-  As claimed in the paper, in order to judge the influence of the filter only from the maximally stretched direction point of view and ignore all other directions, it seems the largest singular value in SVD should be much larger than other singular values. That is, in Figure 1, 에서 Y=FX=(σ_1 u_1 w_1^T )X+(σ_2 u_2 w_2^T )X≈(σ_1 u_1 w_1^T )X seems to hold, but there is no explanation for this.
- This paper proposes a passive filter pruning method, but there is no explanation or comparison about the advantages compared to the active method.
- In the image classification experiment, the MNIST dataset and the VGG16 model alone are insufficient to guarantee the performance of the proposed method.
- In Figure 6, the criteria for determining the pruned layers are not specified, and I am not sure what the meaning of comparing only some layers is pruned.

**Summary Of The Paper:**

In this paper, it is stated that the norm of a convolution filter is insufficient as a criterion for judging how much the filter affects the output of the layer. Instead, the authors state that the operator norm, which indicates how much the filter affects the output during operation, is a good
criterion and can be obtained through SVD.

**Summary Of The Review:**

Although it is a well-written paper, the evidence supporting the proposed technique needs to be clearer. In addition, it is necessary to compare and analyze the performance through additional experiments (refer to the weakness section).

---

> ### Author Response · Authors · 2022-11-19
> **Response to Commnets by Reviewer mkSs (Part1: 1.0)**
>
> ### (Part 1) Summary of Responses Reviewer mkSs (1.0)
>
> We thank reviewer for their valuable suggestions and comments. We appreciate that the reviewer finds that the paper is easy to understand. We have addressed the comments asked by the reviewer as given below,
>
> **Comment 1:** As claimed in the paper, in order to judge the influence of the filter only from the maximally stretched direction point of view and ignore all other directions, it seems the largest singular value in SVD should be much larger than other singular values. That is, in Figure 1, $Y=FX=(\sigma\_1 u\_1 w\_1^T )X+(\sigma_2 u_2 w_2^T )X \approx (\sigma_1 u_1 w_1^T )X$ seems to hold, but there is no explanation for this.
>
> **Response:** In this work, we consider filters as a transformation matrix that transform input feature maps to output feature maps. The operator norm of the filter is equal to the largest singular value ($\sigma_1$) of the filter, and it  represents how maximally the input data gets stretched, when input data is operated by the filter. Also, approximating filter using Rank-1 approximation (which is corresponding to the $\sigma_1$) gives the best least square approximation of the filter.
>
> We analyse  the ratio of $\sigma_1$ and $\sigma_2$ across various networks used for experimentation, and find that $\sigma_1/\sigma_2$ varies from 1.2 to 1.5 across various networks, indicating that the direction corresponding to $\sigma_1$ stretches the input by 1.2 to 1.5 times compared to that of the direction corresponding to $\sigma_2$. In this work, considering direction corresponding to $\sigma_1$ for measuring importance of the filters, appears to be working as the direction represents the best rank-1 approximation and scale the input by 20\% to 50\% higher than that of the direction corresponding to $\sigma_2$.
>
> **Comment 2:** This paper proposes a passive filter pruning method, but there is no explanation or comparison about the advantages compared to the active method.
>
> **Response:** We have revised the manuscript to more clearly compare the proposed passive filter pruning method with active filter pruning method. Please see Page 2 (highlighted text) and Appendix:Table 3 in the revised manuscript.
>
> We use HRank (Lin et al. (2020)), Energy-aware pruning (Yeom et al. (2021)), L1-slimming (Liu et al. (2017)), and GAL-0.1 (Lin et al. (2019)) active filter pruning methods to compare the performance of VGG-16 network using CIFAR-10 dataset.
>
> In general, we find that pruning filters using the proposed pruning method result in a similar reduction of parameters with a drop in accuracy of approximately 1 percentage points compared to the methods that involve dataset to prune filters by using feature maps (Lin et al. (2020); Yeom et al. (2021)), regularization techniques during training (Liu et al. (2017)) or knowledge distillation (Lin et al. (2019)). Therefore, the proposed pruning method is advantageous due to identifying pruned filters without using any dataset, and the pruned network can still achieves accuracy close to that of the active filter pruning methods.
>
> **Comment 3:** In the image classification experiment, the MNIST dataset and the VGG16 model alone are insufficient to guarantee the performance of the proposed method.
>
>
> **Response:**  We have included more experiments on VGG-16 network and ResNet-50 network using CIFAR-10 dataset in the revised manuscript (Appendix). Also, we have updated the code repository covering the  experiments for VGG-16 network and ResNet-50 network on CIFAR-10.
>
> **Comment 4:** In Figure 6, the criteria for determining the pruned layers are not specified, and I am not sure what the meaning of comparing only some layers is pruned.
>
> **Response:** We prune various subset of convolutional layers from the unpruned network. The selection of subset of layers is pseudo-random, where subset of layers are selected in an ordered manner without skipping any convolutional layer.
>
> Please see (Part 2) Summary of Responses Reviewer mkSs (1.0) for remaining response.

---

> > ### Author Response · Authors · 2022-11-19
> > **(Part 2) Summary of Responses Reviewer mkSs (1.0)**
> >
> > ### (Part 2) Summary of Responses Reviewer mkSs (1.0): Part 1 continued...
> >
> > **Comment 5:** This paper used SVD for passive filter pruning without dataset intervention. However, since SVD is already widely used in the pruning field, it seems insufficient in terms of novelty.
> >
> > **Response:** The proposed pruning method performs SVD on filters to identify maximally stretched direction corresponding to operator norm of the filter along which the filter transform the input maximally in order to measure filter importance without involving any dataset. On the other hand, existing methods such as HRank(Lin et al. (2020)) use SVD to compute rank of the features generated using a dataset and energy-aware SVD based pruning method (Yeom et al. (2021)) obtains singular value of feature maps to obtain nuclear norm for defining filter importance using SVD.
> >
> > Other methods which uses SVD are low-rank decomposition or matrix factorization methods (Yu et al. (2017)), which approximate the filter or weight matrix as a low-rank product of two smaller matrices using SVD. In comparison to these methods, the proposed method is a pruning method that eliminate filters from the unpruned CNN, where the number of filters from the unpruned network are changed after pruning that leads to reduce computations and parameters. This is in contrast to the matrix factorization method where the number of filter remains same in the network and computation is reduced by decomposing the filters in low-complexity matrices. It is well studied that low-rank decomposition result in large loss in accuracy under high pruning ratios compared to that of the pruning methods (Lin et al. (2020)).
> >
> > **References:**
> >
> > Mingbao Lin, Rongrong Ji, Yan Wang, Yichen Zhang, Baochang Zhang, Yonghong Tian, and Ling Shao. HRank: Filter pruning using high-rank feature map. In Proceedings of the IEEE/CVF Conference on Computer Vision and Pattern Recognition, pp. 1529–1538, 2020.
> >
> > Shaohui Lin, Rongrong Ji, Chenqian Yan, Baochang Zhang, Liujuan Cao, Qixiang Ye, Feiyue Huang, and David Doermann. Towards optimal structured cnn pruning via generative adversarial learning. In Proceedings of the IEEE Conference on Computer Vision and Pattern Recognition, pp. 2790–2799, 2019.
> >
> > Zhuang Liu, Jianguo Li, Zhiqiang Shen, Gao Huang, Shoumeng Yan, and Changshui Zhang. Learning efficient convolutional networks through network slimming. In Proceedings of the IEEE International Conference on Computer Vision, pp. 2736–2744, 2017.
> >
> > Seul-Ki Yeom, Kyung-Hwan Shim, and Jee-Hyun Hwang. Toward compact deep neural networks via energy-aware pruning. arXiv preprint arXiv:2103.10858, 2021.
> >
> > Xiyu Yu, Tongliang Liu, Xinchao Wang, and Dacheng Tao. On compressing deep models by low rank and sparse decomposition. In Proceedings of the IEEE Conference on Computer Vision and Pattern Recognition, pp. 7370–7379, 2017

---

> > > ### Author Response · Authors · 2022-12-06
> > > **(Part 3) Summary of Responses Reviewer mkSs (1.0)**
> > >
> > > Please follow the link given below for additional experiments on ResNet-50 using Tiny ImageNet dataset and CIFAR-10 dataset,
> > >
> > > https://zenodo.org/record/7406671#.Y4-iBjqnxH5
> > >
> > > For quick summary of the above additional experiments, please follow the section "A brief summary of more experiments after rebuttal suggestions".

---

### Official Review · Reviewer_qjdr · 2022-10-24

**Confidence:** 3
**Correctness:** 3
**Technical Novelty And Significance:** 2
**Empirical Novelty And Significance:** 1
**Recommendation:** 5

**Clarity, Quality, Novelty And Reproducibility:**

- Some pruning methods (Lin et al., 2020; Yeom et al., 2021) apply SVD to features maps to measure the amount of information, but this paper does to weights. It would be helpful to compare these feasure-based and weight-based pruning criteria and discuss pros and cons.
    - Lin et al., HRank: Filter Pruning using High-Rank Feature Map, CVPR'20
    - Yeom et al., Toward Compact Deep Neural Networks via Energy-Aware Pruning, arXiv'21
- The paper is well written and easy to understand.
- I think the empirical validation is too weak and additional experiments with ResNets on large-scale vision datasets are necessary.
- typo: 6p, unrpuned networks at p = 25% -> unpruned


**Strength And Weaknesses:**

Strength
- The proposed method that computes the contribution of filters in producing output is intuitive.
- The experiments for audio-scene classification, which have been less explored in pruning papers, look interesting.
- Figures 1–4 are very helpful for understanding the proposed method; especially, Figure 2 clearly describes the differences between the norm-based filter pruning and the proposed method.
- The ideas in the paper are sufficiently simple for people to build on.

Weakness
- The experimental validation is very weak. For visual classification, the authors performed VGG-on-MNIST experiments solely, which cannot fully support the superiority and applicability of the proposed method. In contrast, recent works (e.g., Molchanov [2019]., Lin [2020], Sui [2021]) always include the experiments with ResNets on ImageNet as well as CIFAR to show the merits of their methods. I would highly recommend the authors to add empirical support on ImageNet and CIFAR with various architectures including ResNets. In particular, how can the proposed method be applied/extended for residual blocks?
- Furthermore, the baselines seem very weak: only the comparison to L1-norm and geometric median pruning (which are classical methods nowadays) was conducted. It would be helpful to include additional recent baselines (e.g., L1 Slimming (Liu et al., 2017), Polarization Regularization (Zhuang et al., 2020), Deep Hoyer (Yang et al., 2019)) to demonstrate the superiority of this work.
- I was not able to identify a clear superiority of the proposed method over the baselines, especially in the VGG-MNIST experiments. In addition, in the DCASE21-DCASE-20 case in Figure 7(b), the results of the operator norm method and those of the GM-based one look very similar at the pruning ratio of 25% and 50%; I think the performance drop at the large pruning ratio (75%, 90%) may be unacceptable and thus the comparison at these ratios may be useless.
- As the authors describe the merits of structured pruning over unstructured pruning include actual inference speedup, it would be good to measure the latency on high-end and/or edge GPUs.


**Summary Of The Paper:**

This paper introduces a weight-based pruning criterion that considers the contribution of the filters in yielding outputs. The importance score is computed using the SVD on the flattened filters and their rank-1 approximation. The experiments are conducted on vanilla convolutional neural networks without residual connections.

**Summary Of The Review:**

The novelty of this work is incremental, while the experimental results are very weak and not impressive.

---

> ### Author Response · Authors · 2022-11-19
> **Response to comments by  Reviewer qjdr Part 1: (1.0)**
>
> ### (Part 1)Summary of Responses Reviewer qjdr (1.0)
> We thank reviewer for their valuable suggestions and comments. We appreciate that the reviewer finds the paper easy to understand. Based on the suggestions from the reviewer, we have performed more experimental analysis and addressed the comments as given below,
>
> **Comment 1:** The experimental validation is very weak. For visual classification, the authors performed VGG-on-MNIST experiments solely, which cannot fully support the superiority and applicability .........add empirical support on ImageNet and CIFAR with various architectures including ResNets. In particular, how can the proposed method be applied/extended for residual blocks?
>
> **Response:**  Reviewer U6V5 also suggested to perform experiments on ImageNet. Given the time to update the manuscript and the resources for running experiments on ImageNet dataset during this phase of rebuttal period, we are able to include experiments on VGG-16 network and a pre-trained ResNet-50 network on ImageNet for CIFAR-10 dataset in the revised manuscript (Appendix). We have updated the code repository (online link is in the revised manuscript) covering the  experiments for VGG-16 network and ResNet-50 network on CIFAR-10. In future, we would like to perform experiments with ImageNet dataset and report the result by the camera-ready submission.
>
> In performing pruning across  ResNet-50 architecture shown in Figure 12 in the revised manuscript (Appendix), we consider convolutional layers having (3 x 3) filters for pruning since they contain more parameters and computations compared to that of the convolutional layers with (1 x 1) filters. The proposed pruning method can  be applied directly to the  convolutional layers in the residual branch as well. For this, the same number of filters from the convolutional layer in the residual branch  should be pruned as that of the convolutional layer from the main "branch c" as shown in Figure 12c in the revised manuscript (Appendix). This is to ensure the same dimensionality while performing addition operation between  the main branch output $F(x)$ and the residual branch output $G(x)$.  After applying the  proposed pruning method on ResNet-50, the number of MACs are reduced by 30\%, the number of parameters are reduced by 33\% at marginal drop in accuracy of 0.29 percentage points compared to the unpruned ResNet-50, as given in Table 4 (Appendix).
>
> **Comment 2:** Furthermore, the baselines seem very weak: only the comparison to L1-norm and geometric median pruning (which are classical methods nowadays) was conducted. It would be helpful to include additional recent baselines (e.g., L1 Slimming (Liu et al., 2017), Polarization Regularization (Zhuang et al., 2020), Deep Hoyer (Yang et al., 2019)) to demonstrate the superiority of this work.
>
> **Response:** The proposed pruning method focuses on pruning filters in a passive manner, where only filters are used to identify set of filters required to retain or prune without involving any dataset. In contrast to the active filter pruning methods, the advantage of the proposed passive pruning method is that there is no need to produce feature maps and perform optimization process while identifying pruned set of filters, which is time-consuming and require more memory resources. Commonly used active filter pruning methods such as HRank (Lin et al. (2020)) and Energy-aware pruning (Yeom et al. (2021)) use feature maps to identify pruned set of filters, L1-slimming (Liu et al. (2017))  and GAL-0.1 (Lin et al. (2019)) use an optimization process to regularize soft mask associated with each feature maps during training process to identify pruned set of filters. In particular, when there is already an existing pre-trained network, identifying pruned set of filters using optimization process (Lin et al. (2019); Liu et al. (2017)) would be heavily computationally expensive compared to that  of the proposed pruning method. Therefore, the proposed work mainly focused on passive filter pruning method.
>
> To analyse the performance of the active and passive filter pruning methods, we include Table 3 in the revised manuscript (Appendix). In general, we find that pruning filters using the proposed pruning method result in a similar reduction of parameters with a drop in accuracy of approximately  1 percentage points compared to the methods that involve dataset to prune filters by using feature maps (Lin et al. (2020); Yeom et al. (2021)), regularization techniques during training (Liu et al. (2017)) or knowledge distillation (Lin et al. (2019)). Therefore, the proposed pruning method is advantageous owing to identifying pruned filters without using any dataset, and the pruned network can still achieves accuracy close to that of the active filter pruning methods. In future, we would like to improve the performance of the proposed framework by improving fine-tuning strategy.
>
>
> Please see Part 2 for other responses.

---

> > ### Author Response · Authors · 2022-11-19
> > **(Part 2)Summary of Responses Reviewer qjdr (1.0)**
> >
> > ### (Part 2) Summary of Responses Reviewer qjdr (1.0), Part 1 continued...
> >
> > **Comment 3:** I was not able to identify a clear superiority of the proposed method over the baselines, especially in the VGG-MNIST experiments. In addition, in the DCASE21-DCASE-20 case in Figure 7(b), the results of the operator norm method and those of the GM-based one look very similar at the pruning ratio of 25\% and 50\%; I think the performance drop at the large pruning ratio (75\%, 90\%) may be unacceptable and thus the comparison at these ratios may be useless.
> >
> > **Response:** We find that the proposed pruning method gives similar performance at smaller pruning ratios and improves performance at high pruning ratio for CNNs designed for audio classification application as shown in Figure 7 in the revised manuscript. In general, the proposed pruning method gives similar or better accuracy at various pruning ratio compared to that of the existing entry-wise norm methods across different CNNs designed for audio and image classification (Figure 7, Figure 10 (Appendix) and Figure 15 (Appendix) in the revised manuscript), hence showing the generalization ability across different domains as well. Therefore, the proposed pruning framework is advantageous over the existing entry-wise norm methods.
> >
> >
> >
> > We agree that the accuracy drop increases as the pruning ratio increases. However, the choice of selecting high pruning ratio may be useful to obtain smaller size CNNs from the large-size CNNs for their deployment on resource-constrained devices having limited computational resources. For example, Cortex-M4 devices (e.g. STM32L496@80MHz or Arduino Nano 33@64MHz) have an upper limit of only 128K on total number of parameters and a maximum limit of only 30M for multiply-accumulate operations (MACs) (Martın-Morato et al. (2022)).
> >
> > Therefore, we believe that it is important to analyse performance at high pruning ratio as well, particularly when we want to deploy CNNs on resource-constrained devices. In this case, the proposed pruning method can be useful compared to that of the existing norm-based pruning methods as the proposed pruning method gives similar or better performance compared to that of the unpruned network.
> >
> > **Comment 4:**  As the authors describe the merits of structured pruning over unstructured pruning include actual inference speedup, it would be good to measure the latency on high-end and/or edge GPUs.
> >
> >
> > **Response:** Thank you very much for your suggestion. In future, we aim to measure latency on high-end or edge GPUs.
> >
> > **Comment 5:** Some pruning methods (Lin et al., 2020; Yeom et al., 2021) apply SVD to features maps to measure the amount of information, but this paper does to weights. It would be helpful to compare these feasure-based and weight-based pruning criteria and discuss pros and cons.
> >     Lin et al., HRank: Filter Pruning using High-Rank Feature Map, CVPR'20 \\
> >     Yeom et al., Toward Compact Deep Neural Networks via Energy-Aware Pruning, arXiv'21
> >
> > **Response:** We have compared with feature map based pruning methods in Table 3 in the revised manuscript. Please refer to response to Comment 2 (Part 1) for detailed explanation.
> >
> >
> > **Comment 6:** I think the empirical validation is too weak and additional experiments with ResNets on large-scale vision datasets are necessary.
> >
> > **Response:** We have performed experiments on VGG-16 and ResNet-50 networks for CIFAR-10 classification and the experimental analysis is included in Appendix in the revised manuscript.
> >
> > **Comment 7:** typo: 6p, unrpuned networks at p = 25\% -> unprune
> >
> > **Response:** In the revised manuscript, we have corrected the mistake.
> >
> > **References:**
> >
> > Mingbao Lin, Rongrong Ji, Yan Wang, Yichen Zhang, Baochang Zhang, Yonghong Tian, and Ling Shao. HRank: Filter pruning using high-rank feature map. In Proceedings of the IEEE/CVF Conference on Computer Vision and Pattern Recognition, pp. 1529–1538, 2020.
> >
> > Shaohui Lin, Rongrong Ji, Chenqian Yan, Baochang Zhang, Liujuan Cao, Qixiang Ye, Feiyue Huang, and David Doermann. Towards optimal structured cnn pruning via generative adversarial learning. In Proceedings of the IEEE Conference on Computer Vision and Pattern Recognition, pp. 2790–2799, 2019.
> >
> > Zhuang Liu, Jianguo Li, Zhiqiang Shen, Gao Huang, Shoumeng Yan, and Changshui Zhang. Learning efficient convolutional networks through network slimming. In Proceedings of the IEEE International Conference on Computer Vision, pp. 2736–2744, 2017.
> >
> > Irene Mart ́ın-Morat ́o, Francesco Paissan, Alberto Ancilotto, Toni Heittola, Annamaria Mesaros,Elisabetta Farella, Alessio Brutti, and Tuomas Virtanen. Low-complexity acoustic scene classification in DCASE 2022 Challenge. In proceedings of DCASE 2022 workshop, 2022.
> >
> > Seul-Ki Yeom, Kyung-Hwan Shim, and Jee-Hyun Hwang. Toward compact deep neural networks via energy-aware pruning. arXiv preprint arXiv:2103.10858, 2021.

---

> > > ### Comment · Reviewer_qjdr · 2022-11-23
> > > **post-rebuttal review**
> > >
> > > I appreciate the authors’ great efforts on addressing the reviewers’ comments. I feel that the additional experimental validation with VGG/ResNet on CIFAR-10 and the comparison with other baselines are good and very helpful. I think measuring latency is optional; however, the experiments on imagenet (which are not reported in the current revised manuscript) are necessary as many other pruning papers presented in top-tier venues have reported them. Therefore, I would like increase my score from 3 (reject, not good enough) to 5 (marginally below the acceptance threshold).

---

> > > > ### Author Response · Authors · 2022-12-06
> > > > **Response to comments by Reviewer qjdr (1.0) after rebuttal**
> > > >
> > > > Thanks for your comments. We are able to perform some additional experiments on Tiny ImageNet dataset. Please see below for the detailed analysis,
> > > >
> > > > For full ImageNet dataset on ResNet-50,  HRank (Lin et al. (2020)) pruning method reduces 36.66\% parameters at an accuracy drop of 1.17 percentage points compared to that of the unpruned ResNet-50. On the other hand,  our pruning method on ResNet-50 using Tiny ImageNet reduces 32\% of parameters at 1.90 percentage points drop in accuracy compared to that of the unpruned ResNet-50. In contrast to pruning ResNet-50 on Tiny ImageNet using HRank method, our method achieve similar accuracy without using any dataset in pruning at reduced computational time in pruning as well. This suggests that our method can achieve similar performance for full ImageNet dataset as well.
> > > >
> > > >
> > > > A summary of new experiments and analysis is explained below,
> > > >
> > > > (Please follow the link for detailed experimental analysis: https://zenodo.org/record/7406671#.Y4-XdjqnxH4.)
> > > > (All figures, Table referenced below are with respect to the above link)
> > > >
> > > >
> > > > ## New experiments
> > > > 1. We perform pruning analysis on ResNet-50 trained on Tiny ImageNet dataset (Wu et al. (2017)). The Tiny ImageNet dataset consists of 200 classes and each class has 500 training examples and 50 validation examples.
> > > >
> > > > 2. Previously, we pruned only the main branch connections each having (3 x 3) filters across different stages in ResNet-50 as shown in Figure 1. Now, we also prune connections, each having (1 x 1) convolutional filters across the residual branch and the main branch in
> > > > different stages of ResNet-50.
> > > >
> > > > 3. We compare the proposed pruning method with feature map based active filter pruning methods such as HRank (Lin et al. (2020)) and Energy-aware (Yeom et al. (2021)) pruning methods. We also use passive filter pruning methods such as entry-wise $l_1$-norm (Li et al.
> > > > (2017)) and entry-wise geometric median (GM) (He et al. (2019)) for comparison.
> > > >
> > > > 4. In addition to performing experimental analysis on ResNet-50 trained using Tiny ImageNet, we also repeat the previous steps (2) and (3) for ResNet-50 trained using CIFAR-10.
> > > >
> > > > ## Summary
> > > >
> > > > (i) For ResNet-50 on Tiny ImageNet as shown in Figure 2, the pruned network obtained after pruning convolutional layers with (3 x 3) filters from stage 4 and stage 5 at 25% pruning ratio results in an accuracy drop of less than 1 percentage point compared to that of the
> > > > unpruned network .
> > > >
> > > > (ii) For ResNet-50 on Tiny ImageNet as shown in Figure 3(a), pruning convolutional layers with (1 x 1) filters along with (3 x 3) filters across various stages at different pruning ratios result in more drop in accuracy compared to that of pruning convolutional layers with (3 x
> > > > 3) filters alone.
> > > >
> > > > (iii) On contrary, for ResNet-50 on CIFAR-10 as shown in Figure 3(b), pruning convolutional layers with (1 x 1) filters along with (3 x 3) filters across various stages at different pruning ratios result similar accuracy with ≤ 0.5 percentage point compared to that of pruning
> > > > convolutional layers with (3 x 3) filters alone.
> > > >
> > > > (iv)  For ResNet-50 on Tiny ImageNet dataset in Table 1, the number of parameters are reduced by 16% at less than 1 percentage points drop in accuracy compared to that of the unpruned network. At 32% reduction in parameters, the accuracy in the pruned network reduces by 1.90 percentage points compared to the unpruned network. For ResNet-50 using CIFAR-10 in Figure 3(b), we find that pruning reduces ≈ 87% parameters at no loss in performance compared to that of the unpruned network.
> > > >
> > > > (v) (ii), (iii) and (iv) suggest that ResNet-50 shows more redundancy for smaller classification problem e.g. CIFAR-10 which has 10 classes and sufficient training example (5000 per class) compared to that of the Tiny ImageNet classification having 200 classes with 500
> > > > examples per class for training. Also, there is a little scope to prune (1 x 1) filters alongwith (3 x 3) filters compared to that of pruning only (3 x 3) filters across various stages in ResNet-50 for Tiny ImageNet dataset due to significant drop in accuracy compared to that
> > > > of the unpruned network as shown in Figure 3(a).
> > > >
> > > > (vi) In contrast to the active filter pruning methods as given in Table 2 for ResNet-50 on Tiny ImageNet, we find that the proposed pruning method is 3 times faster compared to Enegry-aware (Yeom et al. (2021)), 4.5 times faster compared to HRank (Lin et al. (2020)) and gives similar performance without using any dataset in obtaining the pruned network. Similarly, for ResNet-50 on CIFAR-10, the proposed pruning method is 2 times faster compared to Enegry-aware (Yeom et al. (2021)), 3 times faster compared to HRank (Lin et al. (2020)),
> > > > and gives similar performance without using any dataset in obtaining a pruned network. In contrast to the existing passive filter pruning methods, the proposed pruning method achieves similar performance as shown in Figure 4.

---

### Official Review · Reviewer_u6V5 · 2022-10-31

**Confidence:** 3
**Clarity, Quality, Novelty And Reproducibility:** This paper is easy to read, and the n…
**Correctness:** 3
**Technical Novelty And Significance:** 3
**Empirical Novelty And Significance:** 3
**Recommendation:** 6

**Strength And Weaknesses:**

Strength:
1. This paper is easy to read.

2. The authors propose a norm-based filter pruning method.

3. The authors conduct experiments on scene and image classification tasks.

Weaknesses:
1. The novelty of this paper is limited. This paper is based on the norm of channels, and there have been many similar novel papers in recent years.

2. The authors should give experiments with ImageNet dataset.


**Summary Of The Paper:**

In this paper, the authors propose a passive filter pruning method worked on scene classification and image classification tasks. In detail, the authors aim to compute filter importance with the norm of each convolution layer and leverage singular value decomposition to compute a rank-1 approximation of the target channel. The authors provide experiments with TAU Urban Acoustic Scenes 2020 and MNIST datasets.

**Summary Of The Review:**

In this paper, the authors propose a passive filter pruning method worked on scene classification and image classification tasks. In detail, the authors aim to compute filter importance with the norm of each convolution layer and leverage singular value decomposition to compute a rank-1 approximation of the target channel. The authors provide experiments with TAU Urban Acoustic Scenes 2020 and MNIST datasets. The authors should clearly explain the novelty of this paper and conduct experiments with imagenet dataset.

---

> ### Author Response · Authors · 2022-11-19
> **Response to comments by Reviewer u6V5 (1.0)**
>
> ### Summary of responses (1.0: Reviewer u6V5 )
> We thank reviewer for their valuable suggestions and comments. We appreciate that the reviewer finds the paper easy to read. We have  ddressed the comments asked by the reviewer as given below,
>
> **Comment 1**: The novelty of this paper is limited. This paper is based on the norm of channels, and there have been many similar novel papers in recent years.
>
> **Response**: The proposed pruning method uses operator norm of the filter and implicitly considers how a filter transforms a given input to output. This is in contrast to the existing norm-based pruning method which uses entry-wise norm such as absolute sum of the weights (Li et al. (2017)) of the filters. Utilizing operator norm of the filter, our experiments (please see revised manuscript: Figure 7, Figure 10 (Appendix), Figure 15 (Appendix)) across various CNNs designed for audio and image classification reveal that considering operator norm in pruning result in similar or better performance compared to that of the entry-wise norm methods at different pruning ratio. Since, the proposed
> pruning method generalizes better across different domains and hence is useful compared to the existing entry-wise norm methods.
>
> **Comment 2**: The authors should give experiments with ImageNet dataset
>
> **Response**: Given the time to update the manuscript and the resources for running experiments on ImageNet dataset during this phase of rebuttal period, we are able to include experiments on VGG-16 network and a pre-trained ResNet-50 network on ImageNet for CIFAR-10 dataset in the revised manuscript (Appendix). Also, we have updated the code repository (online link is in the revised manuscript) covering the experiments for VGG-16 network and ResNet-50 network on CIFAR-10. In future, we would like to perform experiments with ImageNet dataset and report the result by the camera-ready submission.
>
> Hao Li, Asim Kadav, Igor Durdanovic, Hanan Samet, and Hans Peter Graf. Pruning filters for efficient ConvNets. In International Conference on Learning Representations, 2017.

---

> > ### Comment · Reviewer_u6V5 · 2022-11-25
> > **Review after rebuttal**
> >
> > Thanks for your rebuttal, I think this paper makes some improvements with respect to filter pruning. However, after reading other reviewers' comments, I still feel that the authors should provide more experiments with the ImageNet dataset to support an ICLR paper.

---

> > > ### Author Response · Authors · 2022-12-06
> > > **Response to comments  by Reviewer u6V5 (1.0) after rebuttal**
> > >
> > > Thanks for your comments. We are able to perform experiments on Tiny ImageNet dataset. Please see below for the detailed analysis,
> > >
> > > For full ImageNet dataset on ResNet-50,  HRank (Lin et al. (2020)) pruning method reduces 36.66\% parameters at an accuracy drop of 1.17 percentage points compared to that of the unpruned ResNet-50. On the other hand,  our pruning method on ResNet-50 using Tiny ImageNet reduces 32\% of parameters at 1.90 percentage points drop in accuracy compared to that of the unpruned ResNet-50. In contrast to pruning ResNet-50 on Tiny ImageNet using HRank method, our method achieve similar accuracy without using any dataset in pruning at reduced computational time in pruning as well. This suggests that our method can achieve similar performance for full ImageNet dataset as well.
> > >
> > >
> > > A summary of new experiments and analysis is explained below,
> > >
> > > (Please follow the link for detailed experimental analysis: https://zenodo.org/record/7406671#.Y4-XdjqnxH4.)
> > > (All figures, Table referenced below are with respect to the above link)
> > >
> > >
> > > ## New experiments
> > > 1. We perform pruning analysis on ResNet-50 trained on Tiny ImageNet dataset (Wu et al. (2017)). The Tiny ImageNet dataset consists of 200 classes and each class has 500 training examples and 50 validation examples.
> > >
> > > 2. Previously, we pruned only the main branch connections each having (3 x 3) filters across different stages in ResNet-50 as shown in Figure 1. Now, we also prune connections, each having (1 x 1) convolutional filters across the residual branch and the main branch in
> > > different stages of ResNet-50.
> > >
> > > 3. We compare the proposed pruning method with feature map based active filter pruning methods such as HRank (Lin et al. (2020)) and Energy-aware (Yeom et al. (2021)) pruning methods. We also use passive filter pruning methods such as entry-wise $l_1$-norm (Li et al.
> > > (2017)) and entry-wise geometric median (GM) (He et al. (2019)) for comparison.
> > >
> > > 4. In addition to performing experimental analysis on ResNet-50 trained using Tiny ImageNet, we also repeat the previous steps (2) and (3) for ResNet-50 trained using CIFAR-10.
> > >
> > > ## Summary
> > >
> > > (i) For ResNet-50 on Tiny ImageNet as shown in Figure 2, the pruned network obtained after pruning convolutional layers with (3 x 3) filters from stage 4 and stage 5 at 25% pruning ratio results in an accuracy drop of less than 1 percentage point compared to that of the
> > > unpruned network .
> > >
> > > (ii) For ResNet-50 on Tiny ImageNet as shown in Figure 3(a), pruning convolutional layers with (1 x 1) filters along with (3 x 3) filters across various stages at different pruning ratios result in more drop in accuracy compared to that of pruning convolutional layers with (3 x
> > > 3) filters alone.
> > >
> > > (iii) On contrary, for ResNet-50 on CIFAR-10 as shown in Figure 3(b), pruning convolutional layers with (1 x 1) filters along with (3 x 3) filters across various stages at different pruning ratios result similar accuracy with ≤ 0.5 percentage point compared to that of pruning
> > > convolutional layers with (3 x 3) filters alone.
> > >
> > > (iv)  For ResNet-50 on Tiny ImageNet dataset in Table 1, the number of parameters are reduced by 16% at less than 1 percentage points drop in accuracy compared to that of the unpruned network. At 32% reduction in parameters, the accuracy in the pruned network reduces by 1.90 percentage points compared to the unpruned network. For ResNet-50 using CIFAR-10 in Figure 3(b), we find that pruning reduces ≈ 87% parameters at no loss in performance compared to that of the unpruned network.
> > >
> > > (v) (ii), (iii) and (iv) suggest that ResNet-50 shows more redundancy for smaller classification problem e.g. CIFAR-10 which has 10 classes and sufficient training example (5000 per class) compared to that of the Tiny ImageNet classification having 200 classes with 500
> > > examples per class for training. Also, there is a little scope to prune (1 x 1) filters alongwith (3 x 3) filters compared to that of pruning only (3 x 3) filters across various stages in ResNet-50 for Tiny ImageNet dataset due to significant drop in accuracy compared to that
> > > of the unpruned network as shown in Figure 3(a).
> > >
> > > (vi) In contrast to the active filter pruning methods as given in Table 2 for ResNet-50 on Tiny ImageNet, we find that the proposed pruning method is 3 times faster compared to Enegry-aware (Yeom et al. (2021)), 4.5 times faster compared to HRank (Lin et al. (2020)) and gives similar performance without using any dataset in obtaining the pruned network. Similarly, for ResNet-50 on CIFAR-10, the proposed pruning method is 2 times faster compared to Enegry-aware (Yeom et al. (2021)), 3 times faster compared to HRank (Lin et al. (2020)),
> > > and gives similar performance without using any dataset in obtaining a pruned network. In contrast to the existing passive filter pruning methods, the proposed pruning method achieves similar performance as shown in Figure 4.
> > > .

---

### Author Response · Authors · 2022-11-19
**A brief summary of modification in the revised manuscript**

Based on the suggestions from reviewers in the first discussion stage, we have modified the manuscript with some more experimental analysis. The changes in the revised manuscript w.r.t the initial manuscript are highlighted with a "purple" color.  Also, we have updated the code repository (online link is in the revised manuscript) covering the new experiments. In addition, we include a supplementary material that contains two documents, (1) The revised manuscript without any highlights and (2) Response to comments of all reviewers in a single pdf file.

---

> ### Author Response · Authors · 2022-12-06
> **A brief summary of more experiments after rebuttal suggestions**
>
> Based on the suggestions from the reviewers to perform experiments on ImageNet dataset after modifying the manuscript (revised manuscript), we could perform more experiments on Tiny ImageNet dataset in the given time.
>
> For full ImageNet dataset on ResNet-50,  HRank (Lin et al. (2020)) pruning method reduces 36.66\% parameters at an accuracy drop of 1.17 percentage points compared to that of the unpruned ResNet-50. On the other hand,  our pruning method on ResNet-50 using Tiny ImageNet reduces 32\% of parameters at 1.90 percentage points drop in accuracy compared to that of the unpruned ResNet-50. In contrast to pruning ResNet-50 on Tiny ImageNet using HRank method, our method achieve similar accuracy without using any dataset in pruning at reduced computational time in pruning as well. This suggests that our method can achieve similar performance for full ImageNet dataset as well.
>
>
> A summary of new experiments and analysis is explained below,
>
> (Please follow the link for detailed experimental analysis: https://zenodo.org/record/7406671#.Y4-XdjqnxH4.)
> (All figures, Table referenced below are with respect to the above link)
>
>
> ## New experiments
> 1. We perform pruning analysis on ResNet-50 trained on Tiny ImageNet dataset (Wu et al. (2017)). The Tiny ImageNet dataset consists of 200 classes and each class has 500 training examples and 50 validation examples.
>
> 2. Previously, we pruned only the main branch connections each having (3 x 3) filters across different stages in ResNet-50 as shown in Figure 1. Now, we also prune connections, each having (1 x 1) convolutional filters across the residual branch and the main branch in
> different stages of ResNet-50.
>
> 3. We compare the proposed pruning method with feature map based active filter pruning methods such as HRank (Lin et al. (2020)) and Energy-aware (Yeom et al. (2021)) pruning methods. We also use passive filter pruning methods such as entry-wise $l_1$-norm (Li et al.
> (2017)) and entry-wise geometric median (GM) (He et al. (2019)) for comparison.
>
> 4. In addition to performing experimental analysis on ResNet-50 trained using Tiny ImageNet, we also repeat the previous steps (2) and (3) for ResNet-50 trained using CIFAR-10.
>
> ## Summary
>
> (i) For ResNet-50 on Tiny ImageNet as shown in Figure 2, the pruned network obtained after pruning convolutional layers with (3 x 3) filters from stage 4 and stage 5 at 25% pruning ratio results in an accuracy drop of less than 1 percentage point compared to that of the
> unpruned network .
>
> (ii) For ResNet-50 on Tiny ImageNet as shown in Figure 3(a), pruning convolutional layers with (1 x 1) filters along with (3 x 3) filters across various stages at different pruning ratios result in more drop in accuracy compared to that of pruning convolutional layers with (3 x
> 3) filters alone.
>
> (iii) On contrary, for ResNet-50 on CIFAR-10 as shown in Figure 3(b), pruning convolutional layers with (1 x 1) filters along with (3 x 3) filters across various stages at different pruning ratios result similar accuracy with ≤ 0.5 percentage point compared to that of pruning
> convolutional layers with (3 x 3) filters alone.
>
> (iv)  For ResNet-50 on Tiny ImageNet dataset in Table 1, the number of parameters are reduced by 16% at less than 1 percentage points drop in accuracy compared to that of the unpruned network. At 32% reduction in parameters, the accuracy in the pruned network reduces by 1.90 percentage points compared to the unpruned network. For ResNet-50 using CIFAR-10 in Figure 3(b), we find that pruning reduces ≈ 87% parameters at no loss in performance compared to that of the unpruned network.
>
> (v) (ii), (iii) and (iv) suggest that ResNet-50 shows more redundancy for smaller classification problem e.g. CIFAR-10 which has 10 classes and sufficient training example (5000 per class) compared to that of the Tiny ImageNet classification having 200 classes with 500
> examples per class for training. Also, there is a little scope to prune (1 x 1) filters alongwith (3 x 3) filters compared to that of pruning only (3 x 3) filters across various stages in ResNet-50 for Tiny ImageNet dataset due to significant drop in accuracy compared to that
> of the unpruned network as shown in Figure 3(a).
>
> (vi) In contrast to the active filter pruning methods as given in Table 2 for ResNet-50 on Tiny ImageNet, we find that the proposed pruning method is 3 times faster compared to Enegry-aware (Yeom et al. (2021)), 4.5 times faster compared to HRank (Lin et al. (2020)) and gives similar performance without using any dataset in obtaining the pruned network. Similarly, for ResNet-50 on CIFAR-10, the proposed pruning method is 2 times faster compared to Enegry-aware (Yeom et al. (2021)), 3 times faster compared to HRank (Lin et al. (2020)),
> and gives similar performance without using any dataset in obtaining a pruned network. In contrast to the existing passive filter pruning methods, the proposed pruning method achieves similar performance as shown in Figure 4.
> .

---

### Decision · Program_Chairs · 2023-01-20

**Decision:**

Reject

**Justification For Why Not Higher Score:**

Nearly all authors have pointed out that the authors need to include experiments on imagenet, as this has become a widely accepted experimental setting for pruning works in the literature. During the rebuttal phase, the authors successfully included some extra experimental results on mini-imagenet, which is good. But unfortunately, no reviewers seem to be convinced by this slightly larger experiment. Also, regarding the algorithm itself, the authors suggested entry-wise norm was not a good choice, but why the operator norm could be better, and why it is a good enough indicator of the norm importance? The authors quickly conclude Section 2.1, but did not provide a more in-depth (theoretical) analysis of the proposed indicator.

**Justification For Why Not Lower Score:**

N/A

**Metareview: Summary, Strengths And Weaknesses:**

This paper studied filtering pruning in scene classification and image classification tasks. The authors calculated filter importance through the SVD on the flattened filters and their rank-1 approximation. The algorithm has been introduced in an intuitive way with a few carefully designed figures. The major concern lies in the experimental part.